# USDC: A Dataset of User Stance and Dogmatism in Long Conversations

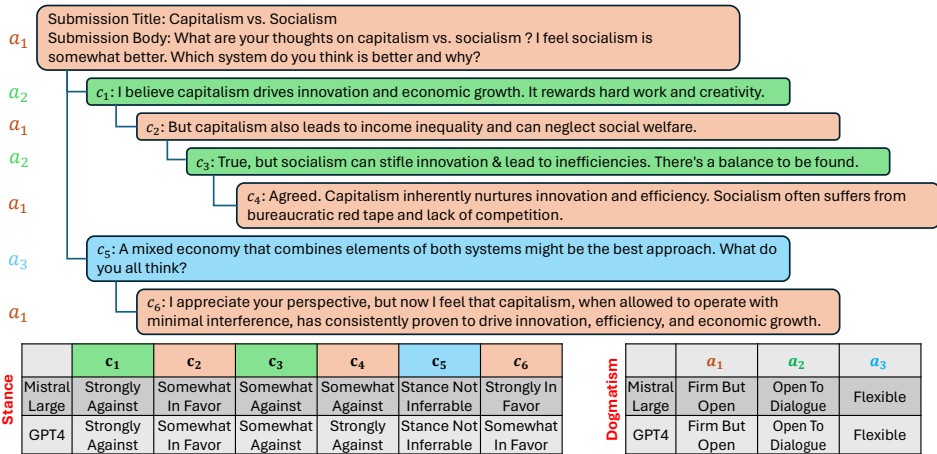

Figure 1: Sample Reddit conversation on "Capitalism vs. Socialism" with Stance (for every comment $\{c_i\}_{i=1}^{6}$) and Dogmatism (for every author $\{a_j\}_{j=1}^{3}$) labels from Mistral Large and GPT-4. The submission content favors to socialism and examines how the authors position their opinions regarding socialism vs. capitalism.

## Abstract

Identifying user's opinions and stances in long conversation threads on various topics can be extremely critical for enhanced personalization, market research, political campaigns, customer service, conflict resolution, targeted advertising and content moderation. Hence, training language models to automate this task is critical. However, to train such models, gathering manual annotations has multiple challenges: 1) It is time-consuming and costly; 2) Conversation threads could be very long, increasing chances of noisy annotations; and 3) Interpreting instances where a user changes their opinion within a conversation is difficult because often such transitions are subtle and not expressed explicitly. Inspired by the recent success of large language models (LLMs) for complex natural language processing (NLP) tasks, we leverage Mistral Large and GPT-4 to automate the human annotation process on the following two tasks while also providing reasoning: i) User Stance classification, which involves labeling a user's Stance of a post in a conversation on a five-point scale; ii) User Dogmatism classification, which deals with labeling a user's overall opinion in the conversation on a four-point scale. The majority voting on zero-shot, one-shot, and few-shot annotations from these two LLMs on 764 multi-user Reddit conversations helps us curate the USDC dataset. USDC is then used to finetune and instruction-tune multiple deployable small

language models for the 5-class stance and 4-class dogmatism classification tasks.
We make the code and dataset publicly available [1].

.

# 1 Introduction

Understanding the user's (or author's) opinion in a conversation is a fundamental aspect of successful interpersonal interactions, and it is essential for developing better interpersonal communication skills, empathy development, and informed decision-making. This user understanding is particularly relevant in the context of dogmatism, a phenomenon observed in various areas such as politics, religion, culture, intellect, and science, where rigid adherence to beliefs often hinders open-mindedness and empathy (Rokeach, 1954). Advertisers can target their campaigns more effectively by aligning with the opinions and stances of potential customers. Companies can use this information for market research to tailor products and services to meet consumer needs and preferences. Political groups can gauge public reaction to policies and campaigns and adjust their strategies accordingly. Identifying differing opinions can help conflict resolution by understanding the perspectives of all parties' perspectives. Society can promote tolerance and maintain social harmony by recognizing and respecting diverse opinions.

Fig. 1 shows a sample Reddit conversation on the topic of *Capitalism vs. Socialism*. We refer to an author's initial post (containing title and body) as a submission. Multiple authors can then share their opinions as comments on the submission. Specifically this example contains 6 comments $\{c_i\}_{i=1}^6$ from 3 authors $\{a_j\}_{j=1}^3$. We also show stance and dogmatism predictions from two large language models (LLMs): Mistral Large and GPT-4. Some authors like $a_1$ change their views during the discussion based on the beliefs or opinions of others. At the beginning of the dialogue, we note that author $a_1$ is somewhat favoring socialism (in submission and $c_2$). But the author shifts their stance to somewhat favors capitalism (in $c_4$) after considering the viewpoints of author $a_2$ in comments $c_1$ and $c_3$, illustrating author $a_1$'s firm yet open-minded approach. On the other hand, author $a_3$ seems very flexible based on their comment $c_5$. Understanding conversations requires understanding the fine-grained topics being discussed and the dynamic viewpoints of the individual users.

Given the importance of understanding these user dynamics in conversations, training language models to perform this task automatically at scale is critical. While numerous datasets are available for analyzing individual user posts (Fast & Horvitz, 2016; Sakketou et al., 2022), typically through random subsampling of posts or selecting posts with a limited number of tokens, the exploration of a specific user's opinion across each post within an entire conversational thread remains under-explored.

Crowdsourcing is one possible approach to address the need for a suitable dataset. However, a significant limitation in manually annotating datasets for user opinions is the time-consuming nature of the process, as annotators must read entire conversations to label each user's post, making data acquisition costly. Additionally, manual annotation often faces challenges related to quality, as accurately labeling opinions requires understanding demographic details and domain-specific knowledge. Given these limitations, achieving a comprehensive and accurate set of user opinions corresponding to posts about a topic often requires multiple annotators or iterative rounds of annotation. Since users could change their opinion (often times with subtle transitions and not with explicit statements) within a conversation, tracking such changes across multiple users manually becomes very cumbersome.

Recently, large language models (LLMs), especially those built on Transformer architectures (Vaswani et al., 2017) and pretrained on large datasets, have resulted in state-of-the-art accuracies on several complex natural language processing (NLP) tasks (Brown et al., 2020; Chung et al., 2024). LLMs are also being frequently used for dialog response generation (Zhang et al., 2020; Bao et al., 2019; Roller et al., 2021; Adiwardana et al., 2020). Given the complex and cumbersome nature of conversation understanding, we hypothesize that LLMs can be effective in capturing nuances involved in understanding user opinions and their shifts in multi-user conversational contexts. Also, since these models possess long-range memory capabilities, we believe that they can reason over extended conversational threads involving numerous participants, as good as human annotators, if not better.

In this work, we leverage LLMs like Mistral Large and GPT-4 to perform two tasks: i) User Stance classification, which involves labeling a user's stance of a post in a conversation on a five-point

---

[1] https://anonymous.4open.science/r/USDC-0F7F

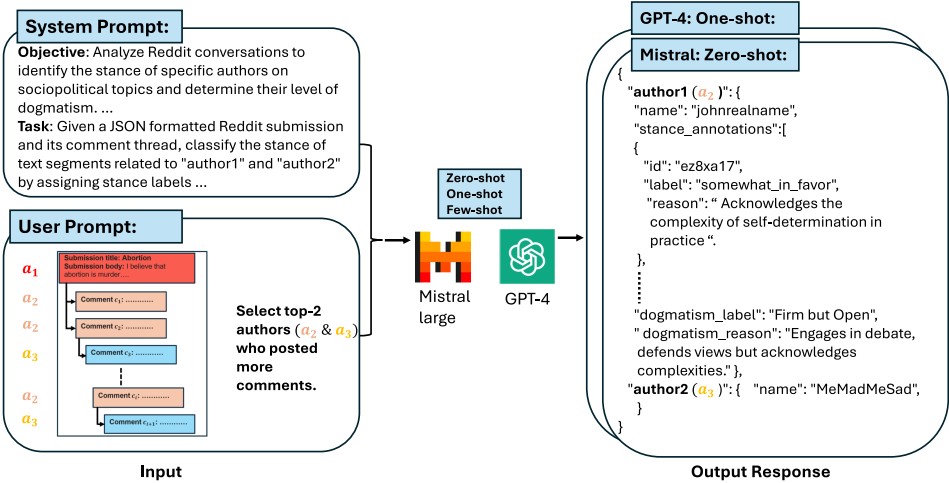

Figure 2: Generating annotations using LLMs: We pass the entire conversation for each Reddit thread in JSON format. The JSON highlights the top two authors who posted the most comments, alongside annotation guidelines for stance and dogmatism labels in the system prompt.

scale; ii) User Dogmatism classification, which deals with labeling a user's overall opinion in the conversation on a four-point scale. Besides the class labels, we also obtain reasoning behind these labels from these LLMs. We experiment with these two models as human-like annotators to generate user opinions in full-length, multi-user Reddit conversations in a zero-shot, one-shot as well as few-shot setup. Thus, overall for every sample, we obtain six annotations ({Mistral Large, GPT-4}×{zero-shot, one-shot, few-shot}). Fig. 2 presents our LLM-based annotation pipeline for user-level Stance and Dogmatism tasks. We consider majority voting over these six as our final annotations. Overall, this helps us curate our USDC (a dataset of user stance and dogmatism in conversations) dataset, which consists of 764 multi-user conversations from 22 subreddits, including 1,528 user-level dogmatism samples and 9,618 stance samples across all posts from selected users. Overall, the annotations in the dataset highlight specific user opinions in each post related to stance, track opinion fluctuations leading to a dogmatic nature, and provide reasoning about why users hold specific opinions.

USDC addresses several weaknesses of existing post-level stance and dogmatism datasets. First, the full-length multi-user conversation aspect of USDC enables it to capture contextual and opinion shifts of multiple users. This feature allows it to serve as both an instruction-tuning user opinion dataset and an evaluation benchmark. We believe that the ability to perform instruction tuning for user opinions at a large scale can bridge the gap between open-source and commercial user trait understanding models. Additionally, the in-context learning annotations using state-of-the-art LLMs in USDC make it a more comprehensive measure of how current LLMs understand complex tasks like capturing opinions. This aspect makes it a valuable resource, especially for social media agents seeking deeper insights into user behavior.

In this work, we utilize our USDC dataset to finetune as well as instruction-tune open-source LLMs for generating stance and dogmatism labels for users. We experiment with three pretrained small language models (SLMs) like LLaMA-2-7B, LLaMA-3-8B, and Falcon-7B. We also experiment with four instruction-tuned SLMs like LLaMA-2-chat-7B, LLaMA-3-8B-instruct, Vicuna-7B-v.1.5, and Falcon-7B-instruct. We report weighted F1 scores obtained using these models for both the tasks.

We make the following contributions: 1) We contribute USDC (a dataset of user stance and dogmatism in conversations) dataset consisting of 764 multi-user conversations labeled with 1,528 user-level dogmatism samples and 9,618 stance samples. 2) We report initial results for the stance and dogmatism detection tasks using seven small language models for the UDSC dataset. We find that stance detection performance improves with instruction-tuning (F1-score of 56.2) compared to fine-tuning (F1-score of 54.9). However, dogmatism detection performs worse with instruction-tuning (F1-score of 49.2) compared to fine-tuning (F1-score of 51.4), highlighting the complexity of this task. 3) We make the code and dataset publicly available[1]. Also, the finetuned and instruction-tuned models are made available as well.

## 2  Related Work

**Opinion fluctuations in user conversations.** Our work is closely related to previous studies (Fast & Horvitz, 2016; Sakketou et al., 2022), which explore Stance and Dogmatism at the post level, where posts are randomly sampled from conversation threads. Fast & Horvitz (2016) predicted user dogmatism on randomly sampled Reddit posts from conversations, with each post limited to 200-300 characters. One major limitation of this work is the unavailability of a public dataset and missing annotator demographic details. Sakketou et al. (2022) created the post-level Stance dataset, SPINOS, where each post is considered independent, and submission posts are missing while annotators label the data. Additionally, the quality of the dataset is not validated due to missing demographic details of these annotators. Our work overcomes the limitations of previous studies and presents Stance detection for posts and Dogmatism labels of users in conversations, considering the entire context, while preserving submission IDs. Hence, our dataset provides clear user-level posts and dogmatism data, which are useful for modeling dynamic user representations.

**Generating annotations for NLP tasks using Large Language Models** Our work also relates to a growing body of literature suggesting that large language models can perform similarly to human annotators in labeling complex NLP tasks (Zhou et al., 2022; Zhang et al., 2023; Bansal & Sharma, 2023; Lowmanstone et al., 2023; Wadhwa et al., 2023; Honovich et al., 2023; Zheng et al., 2024; Ye et al., 2022a; Meng et al., 2022). Several studies have explored LLM-based annotation generation in zero-shot or few-shot task settings (Ye et al., 2022a; Meng et al., 2022; Ye et al., 2022b), while others have compared pairs of language models to assess the quality of annotations generated by these LLMs (Zheng et al., 2024). However, these studies focused on generating annotations for NLP tasks such as sentiment analysis, natural language inference (Gilardi et al., 2023; Alizadeh et al., 2023), or creating synthetic dialogues, but only for dyadic conversations (Lee et al., 2023). Our approach complements these previous studies by focusing on generating annotations of user opinions in complex multi-user conversations.

## 3  USDC Dataset Curation

In this section, we will discuss three main things: 1) Collection of Reddit conversations, 2) Obtaining LLM annotations, and 3) Inter-annotator agreement with LLMs as annotators.

### 3.1  Collection of Reddit Conversation Threads

**Initial crawl.** We crawl an year (2022) worth of multi-user conversation data from 22 subreddits of Reddit [2] using praw API [3]. This dataset includes submissions and all associated user comments. Each submission, which serves as the initial message of the conversation, contains a title and content body. This is followed by comments and replies to the submission or other comments. Overall, we crawled 3,619 Reddit conversations across the 22 subreddits. A sample Reddit conversation is displayed in Fig. 1.

**Quality filtering of conversations.** Since submission content on Reddit can sometimes include videos, we perform the following filtering steps. 1) We only consider submissions where the content is text. 2) We remove conversations with [deleted] tags and empty content. 3) We exclude conversations where the posts were either discarded by users or removed by moderators.

Reddit user conversations can be very long and we observed up to 591 comments in a single crawled conversation data. Considering the maximum sequence length allowed by various language models, we retained only those conversations that contain at least 20 and at most 70 comments. Considering conversations with fewer than 20 comments results in too few comments to accurately gauge user opinions based on small samples. Further, we ensure that at least two users covering ~50% of the comments in the conversations. We did not remove any comments or reduce the post length in the selected conversations. Out of the initial 3,619 conversations, these filtering steps result into 764 conversations getting selected. Table. 4 in the Appendix shows detailed subreddit level statistics.

---

[2]https://www.reddit.com/
[3]https://github.com/praw-dev/praw

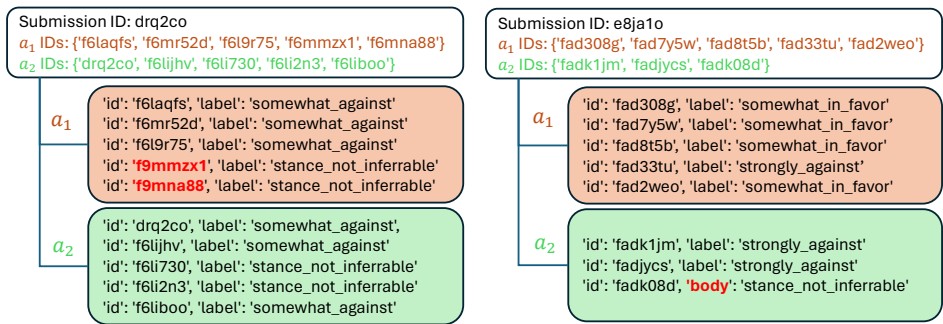

Figure 3: Failure cases of LLMs: Mistral Large few-shot output (left), the ids ("f6mmzx1","f6mna88") were mismatched with generated ids ("f9mmzx1","f9mna88"), GPT-4 zero-shot output (right), the key **"label"** was mismatched with generated key "body".

## 3.2 Obtaining LLM Annotations

**Representing Reddit conversations in JSON format.**

To create the prompt, we follow the nested hierarchical structure of Reddit conversations to maintain the context. Specifically, we maintain a JSON structure for each conversation, where each author has their post IDs, and comments or replies are available in the body section. An example of a Reddit conversation in JSON format is provided in Appendix D. Note that the JSON explicitly includes the top-2 authors who posted the most comments in the conversation, as well as their respective post IDs. Our emphasis on these top-2 users (covering 47% posts of total posts on average) aimed at accurately assigning Stance and Dogmatism labels, acknowledging the challenge of modeling a user's opinion belief based on a very number of posts within a conversation.

**Using LLMs as human-like annotators.** To annotate the position (or Stance) of a user towards a subreddit topic at each post and the overall opinion (or Dogmatism level) of a user in a conversation, we employ two well-known commercialized API-based LLMs: GPT-4 (OpenAI, 2023) and Mistral Large (Jiang et al., 2024). OpenAI GPT-4 is a decoder-based language model which features a context window of 32k to 128k tokens. Mistral Large features a context window of 32k tokens. Additionally, we also examined other versions of these models, such as GPT-3.5 and Mistral-small and medium, but found that these models failed to produce annotations in the desired format. We briefly discuss these limitations in Section 6.

For both GPT-4 and Mistral Large, we supplied a system prompt that contains the definition of Stance and Dogmatism, guidelines for annotating each user conversation, and the necessary labels for Stance and Dogmatism, as shown in Fig 2. The system prompt is detailed in the Appendix B. Along with the system prompt, we provided a user prompt comprising the entire user conversation in a structured JSON format, as discussed above. Additionally, we prompted the model to generate reasoning for each label, explaining why the LLMs assigned a particular label to a specific user post. We used zero-shot, one-shot, and few-shot settings to get the LLM-based annotations. For the few-shot setting, we added two examples in the prompt. Samples of generated outputs using GPT-4 in zero-shot, one-shot, and few-shot settings are shown in Appendix E.1, E.2, E.3 respectively. Similarly, samples of generated outputs using Mistral Large in zero-shot, one-shot, and few-shot settings are shown in Appendix E.4, E.5, E.6 respectively.

**Annotation tasks.** We prompt the LLMs to perform two annotation tasks: 1) Stance detection, which determines if a user comment or post is *Strongly In Favor*, *Strongly Against*, *Stance Not Inferrable*, *Somewhat In Favor*, or *Somewhat Against* towards specific subreddit submission content; 2) Dogmatism identification, which evaluates the user's overall opinion in conversation and categorizes them into one of four categories: *Firm but Open*, *Open to Dialogue*, *Flexible* or *Deeply Rooted*. This assessment reveals whether a user is open to changing their beliefs or remains steadfast in their opinions based on interactions with other users.

**Addressing LLM response and JSON parsing failures.** Sometimes the LLMs got confused with the author IDs and missed Stance labels for some author IDs (Fig. 3 (left)). Sometimes, there were minor errors in key naming ('label' vs 'body' in Fig. 3 (right)). For each LLM setting, we observed

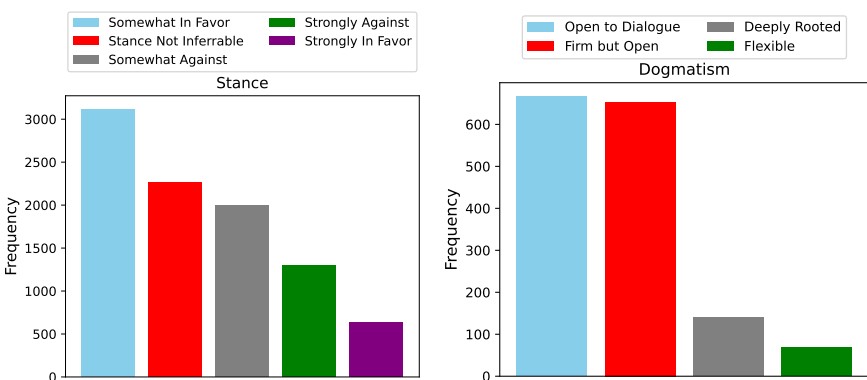

Figure 4: Distribution of class labels for Stance (left) and Dogmatism (right) tasks. These class labels are determined by majority voting across GPT-4 and Mistral Large models.

such errors in around 15 cases on average. We manually fixed such JSON parse errors and missing Stance labels for some author IDs.

**Majority voting.** After obtaining six annotations ({Mistral Large, GPT-4}×{zero-shot, one-shot, few-shot}) for each sample, we aggregate using majority voting to determine the final gold annotations for the Stance and Dogmatism tasks. Fig. 4 presents the class distributions for both the annotation tasks. Additionally, we present the class distributions obtained from each model with the three settings (zero-shot, one-shot and few-shot) for two tasks in Appendix Figs. 5 and 6 respectively.

### 3.3   Inter-annotator Agreement with LLMs as Annotators

As the quality of labeling on subjective tasks is challenging, we validated the inter-annotator agreement (IAA) between the six LLMs (GPT-4 Zero-shot, GPT-4 One-shot, GPT-4 Few-shot, Mistral Large Zero-shot, Mistral Large One-shot, and Mistral Large Few-shot) for the Stance as well as Dogmatism tasks. We perform IAA using two approaches: i) Cohen's kappa score (Cohen, 1960) and ii) Fleiss' kappa score (Fleiss, 1971). Cohen's kappa measures the agreement between two raters, while Fleiss' kappa extends this to multiple raters. Hence, we employed Cohen's kappa for pairwise comparisons and Fleiss' kappa for overall agreement across all models.

Fig. 7 in Appendix shows the pairwise Cohen's kappa values for both Stance and Dogmatism tasks. We observe that Cohen's kappa values range from 0.36 to 0.72 for Stance and 0.31 to 0.61 for dogmatism, indicating moderate agreement between the models. Broadly kappa values are higher for model pairs within a family (GPT-4 or Mistral large). Thus, the large variance in the kappa scores is not due to the various in-context learning settings (ZS, OS, FS) but rather due to architectural differences.

The overall Fleiss' kappa value was calculated as 0.485 for Stance and 0.435 for Dogmatism, suggesting moderate agreement among all six models. Comparing LLM IAA with previous studies, we observe that for dogmatism, the LLM IAA of 0.435 matches with 0.44 as mentioned in Fast & Horvitz (2016). Similarly, for Stance, the LLM IAA of 0.485 is much higher than 0.34 as reported in Sakketou et al. (2022). It is important to note that previous studies on Stance and Dogmatism datasets were created on post-level data with limited token lengths, whereas our work focuses on entire user conversations. This suggests that LLMs can be considered as competent annotators for complex subjective tasks. However, the moderate agreement levels indicate potential areas for improvement and align with the observed performance variations among the models.

## 4   Training Small Language Models

In this section, we briefly discuss the small language models that we experiment with. We also discuss their finetuning and instruction tuning details.

### 4.1 Small Language Models

we train three pretrained small language models (LLaMA-2-7B, LLaMA-3-8B, Falcon-7B) and four instruction-tuned small language models (LLaMA-2-chat-7B, LLaMA-3-8B-instruct, Vicuna-7B-v.1.5, and Falcon-7B-instruct). We finetune as well as instruction tune these models using the proposed USDC dataset. We use pretrained models checkpoints from Hugging Face. All of these LLMs have context length of 4096 tokens.

**LLaMA** models (Touvron et al., 2023a) are decoder-only LLMs trained on 1.6 trillion tokens from a mixture of corpora including C4, English CommonCrawl, Wikipedia, Github, and more. We use two versions of models in our study: LLaMa-2-7B (Touvron et al., 2023b) and LLaMa-3-8B and their instruction tuned variants.

**Falcon** models (Almazrouei et al., 2023) are decoder-only LLMs trained on $\geq 1$ trillion tokens of text, with a particular emphasis on the RefinedWeb corpus. For Falcon, we use both the pretrained and instruction tuned 7B parameter variants in our study.

**Vicuna** model (Chiang et al., 2023) is finetuned from the LLaMA 7B model on approximately 70K user-shared conversations gathered from ShareGPT.com and we used the 7B parameter variants.

### 4.2 Experimental Setup

**Train-test setup.** We conducted both finetuning and instruction-tuning of small language models. For this purpose, we divided the dataset of 764 conversations into train ($\sim 75\%$) and test splits ($\sim 25\%$). The training dataset comprised 564 conversations, including 1128 samples of Dogmatism labels and 7520 samples of Stance labels. Conversely, the testing dataset consisted of 200 conversations, with 400 samples of Dogmatism labels and 1831 samples of Stance labels across two authors posts.

**Implementation details for reproducibility.** All experiments were conducted on a machine equipped with an NVIDIA A100 GPU with 80 GB of GPU RAM, partitioned into two devices of 40 GB each. We employed 4-bit quantization with normalized floating precision (nf4) from the bitsandbytes library [4]. Additionally, we utilized LoRA (Hu et al., 2021) with a rank of 64 and an alpha value of 16 during task-based instruction tuning. Finally, we use PEFT (Parameter Efficient Finetuning) [5] library to train large language models with SFTT (Supervised Finetuning Trainer) setting. To further enhance performance, we divided the training dataset into a validation set, comprising a randomly chosen 10% subset from the training set, used exclusively for hyperparameter tuning. More details about bitsandbytes, PEFT and SFTT parameters are reported in Appendix.

### 4.3 Finetuning and Instruction Tuning of Small Language Models (SLMs)

**Finetuning of SLMs.** For Stance classification, we treat each user post as an independent sample. In contrast, for Dogmatism classification, we consider the entire user conversation as a single sample by concatenating all the threads from a user in that conversation. To load the pretrained SLMs, we perform 4-bit quantization, apply the LoRA technique (Hu et al., 2021), and fine-tune the models with SFTT before saving the fine-tuned model. For finetuning, we used prompt for Stance classification as shown in Fig. 8 (see Appendix). Similarly, Fig. 9 (see Appendix) displays prompt for Dogmatism identification.

**Instruction tuning of SLMs.** We instruction tune the SLMs on user conversations along with their gold labels from the training part of the USDC dataset. For instruction tuning, we use the same prompt as used for LLMs to generate the USDC dataset (also shown in Appendix B). Similar to finetuning, we use same train-test splits for instruction tuning.

## 5 Results

**Do SLMs finetuned with task-specific LLM annotations accurately perform Stance and Dogmatism tasks on user opinions?** We show the weighted F1 of various SLMs finetuned with task-specific LLM annotations on the stance and dogmatism detection tasks on the USDC test set in Table 1. We

---

[4] https://pypi.org/project/bitsandbytes/
[5] https://github.com/huggingface/peft

Table 1: Finetuning: weighted F1 score for Stance classification using SLMs on USDC test set. ZS: Zero-shot, OS: One-shot, FS: Few-shot.

| Model | Stance Classification | | | | | | | Dogmatism Classification | | | | | | |
| | GPT-4 | | | Mistral Large | | | Majority | GPT-4 | | | Mistral Large | | | Majority |
| | ZS | OS | FS | ZS | OS | FS | | ZS | OS | FS | ZS | OS | FS | |
|---|---|---|---|---|---|---|---|---|---|---|---|---|---|---|
| LLaMA-2-7B | 51.8 | 52.9 | 52.7 | 35.1 | 49.2 | 46.0 | 54.0 | 42.1 | 44.2 | 45.2 | 39.3 | 47.6 | 43.7 | 43.4 |
| LLaMA-2-chat-7B | 52.8 | 51.4 | 51.8 | 34.7 | 47.5 | 46.5 | 51.3 | 42.1 | 42.5 | 48.8 | 41.1 | 49.7 | 45.5 | 48.3 |
| LLaMA-3-8B | 51.3 | 52.2 | 52.9 | 34.9 | 48.5 | 47.0 | **54.9** | 42.0 | 47.8 | 45.3 | 39.9 | 47.4 | 36.3 | **51.4** |
| LLaMA-3-8B-instruct | 51.2 | 52.6 | 52.7 | 33.9 | 49.5 | 45.6 | 54.5 | 44.8 | 46.2 | 49.7 | 46.1 | 45.8 | 46.1 | 50.8 |
| Falcon-7B | 50.7 | 51.1 | 51.6 | 34.9 | 47.2 | 43.9 | 53.2 | 41.5 | 42.1 | 43.3 | 36.5 | 38.4 | 37.5 | 40.1 |
| Falcon-7B-instruct | 51.2 | 51.5 | 51.6 | 35.1 | 47.7 | 44.2 | 51.0 | 41.7 | 42.1 | 42.9 | 36.8 | 38.5 | 36.9 | 39.7 |
| Vicuna-7B-v.1.5 | 51.0 | 53.0 | 53.2 | 35.1 | 48.5 | 45.8 | 54.7 | 42.9 | 48.3 | 40.8 | 45.9 | 42.6 | 46.2 | 42.3 |

Table 2: Instruction-tuning: weighted F1 score for Stance classification using SLMs on USDC test set. ZS: Zero-shot, OS: One-shot, FS: Few-shot.

| Model | Stance Classification | | | | | | | Dogmatism Classification | | | | | | |
| | GPT-4 | | | Mistral Large | | | Majority | GPT-4 | | | Mistral Large | | | Majority |
| | ZS | OS | FS | ZS | OS | FS | | ZS | OS | FS | ZS | OS | FS | |
|---|---|---|---|---|---|---|---|---|---|---|---|---|---|---|
| LLaMA-2-7B | 53.2 | 54.0 | 54.5 | 36.8 | 50.3 | 47.2 | 55.5 | 43.0 | 45.0 | 46.3 | 40.6 | 48.2 | 45.0 | 44.0 |
| LLaMA-2-chat-7B | 54.0 | 54.5 | 55.0 | 36.5 | 50.7 | 47.6 | 54.0 | 43.2 | 45.5 | 47.0 | 40.8 | 48.5 | 45.5 | 43.8 |
| LLaMA-3-8B | 53.5 | 54.8 | 55.5 | 37.0 | 50.5 | 48.0 | **56.2** | 43.5 | 46.0 | 47.5 | 41.0 | 48.8 | 45.8 | 45.1 |
| LLaMA-3-8B-instruct | 53.0 | 54.2 | 55.0 | 36.0 | 50.0 | 47.0 | 55.5 | 43.8 | 46.5 | 47.8 | 41.5 | **49.2** | 46.0 | 44.8 |
| Falcon-7B | 52.8 | 53.4 | 54.0 | 36.5 | 49.5 | 46.5 | 54.8 | 42.5 | 44.6 | 45.8 | 39.8 | 47.0 | 44.0 | 43.8 |
| Falcon-7B-instruct | 53.0 | 53.8 | 54.2 | 36.8 | 49.8 | 46.8 | 54.5 | 42.8 | 44.8 | 46.0 | 40.0 | 47.2 | 44.2 | 43.0 |
| Vicuna-7B-v.1.5 | 53.3 | 54.5 | 55.2 | 37.0 | 50.2 | 47.8 | 55.2 | 43.7 | 46.8 | 47.2 | 41.2 | 48.2 | 46.5 | 44.8 |

report AUC scores and other qualitative analysis in Appendix F (Fig. 11 and 12). We make the following observations from these results: 1) For both tasks, the majority voting labels as ground truth, has a relatively high performance, scoring above 50% weighted F1-score across several models. 2) LLaMa-3 models (LLaMA-3-8B and LLaMA-3-8B-instruct) perform better across both the tasks. 3) For GPT-4 annotations, in most cases, SLMs finetuned with few-shot annotations outperform those trained with zero and one-shot annotations. For Mistral Large annotations, typically SLMs finetuned with one-shot annotations performs the best. 4) Specifically, for Stance detection task, Vicuna-7B-v.1.5 finetuned using few-shot annotations is the best model trained with GPT-4 annotations. Similarly, LLaMA-3-8B-instruct finetuned with one-shot annotations is the best model trained with Mistral Large annotations. 5) For the Dogmatism detection task, LLaMA-3-8B-instruct finetuned using few-shot annotations is the best model trained with GPT-4 annotations. Similarly, LLaMA-2-chat-7B finetuned with one-shot annotations is the best model trained with Mistral Large annotations. 6) Overall, we observe that instruction tuned SLMs perform better than the pretrained SLMs.

**Do SLMs instruction-tuned with task-specific LLM annotations perform better than SLMs finetuned with task-specific LLM annotations for the Stance and Dogmatism tasks?** We show the weighted F1 of various SLMs instruction-tuned with task-specific LLM annotations, on the stance and dogmatism detection tasks on the USDC test set in Table 2. We report AUC scores and other qualitative analysis in Appendix F (see Fig. 13). We make the following observations from these results: 1) SLMs with instruction-tuning result in higher weighted F1-scores than SLMs with finetuning for stance detection, while SLMs with finetuning outperform SLMs with instruction-tuning in dogmatism detection. 2) Contrary to results in Table 1, Table 2 demonstrates that using majority voting labels as ground truth, SLM instruction-tuning yields relatively high performance only for the stance detection task, but not for the dogmatism detection. 3) Similar to results in Table 1, LLaMA-3 models (LLaMA-3-8B and LLaMA-3-8B-instruct) perform better across both tasks. Additionally, GPT-4 annotations yield the best results in the few-shot setting, while Mistral Large annotations perform best in the one-shot setting.

Overall, we draw the following conclusions when comparing SLM finetuning and instruction-tuning: (1) Since dogmatism detection is inherently a more complex and varied task than stance detection, the model might struggle to generalize from the instructional data. (2) The system prompt used in finetuning is much simpler than the original system prompt for instruction-tuning, making it challenging to handle the context length for longer conversations. We perform an error analysis to further analyze the results in the next subsection.

**Error Analysis** Table 3 illustrates the confusion matrix for stance detection for LLaMa-3-8B finetuning and instruction-tuning. We make the following observations this table: 1) For both finetuning and instruction-tuning, there is a significant misclassification between "Somewhat Against" and "Somewhat In Favor," as well as between "Somewhat In Favor" and "Stance Not Inferrable." These overlaps suggest challenges in distinguishing moderate stances, indicating a need for enhanced

| | | Predicted | | | | | | | | Predicted | | | | |
|---|---|---|---|---|---|---|---|---|---|---|---|---|---|---|
| | | SOA | SOIF | SNI | SGA | SIF | | | SOA | SOIF | SNI | SGA | SIF |
| | SOA | 151 | 132 | 34 | 44 | 2 | | SOA | 143 | 125 | 37 | 54 | 4 |
| | SOIF | 93 | 537 | 113 | 17 | 14 | | SOIF | 82 | 543 | 106 | 27 | 16 |
| Actual | SNI | 23 | 78 | 259 | 5 | 0 | Actual | SNI | 22 | 82 | 253 | 6 | 2 |
| | SGA | 52 | 35 | 13 | 115 | 17 | | SGA | 41 | 35 | 11 | 131 | 14 |
| | SIF | 18 | 50 | 12 | 25 | 27 | | SIF | 16 | 53 | 10 | 23 | 30 |

Table 3: Confusion matrix for LLaMa-3-8B Stance detection models on USDC test set: finetuning (left) and instruction-tuning (right). SOA: Somewhat Against, SOIF: Somewhat In Favor, SNI: Stance Not Inferrable, SGA: Strongly Against, SIF: Strongly In Favor.

feature representation and clearer class definitions to improve model performance. We report the confusion matrix for dogmatism detection task in Appendix Fig. 10. Fig. 10 shows significant misclassifications, especially for the "Deeply Rooted" and "Flexible" labels, with both having zero accuracy and F1-scores. On the other hand, the model performs moderately better for "Firm but Open" and "Open to Dialogue" classes with accuracies of 48.7% and 64.4% respectively. The confusion matrix also indicates substantial confusion to distinguish between intermediate levels of dogmatism, such as "Firm but Open" and "Open to Dialogue". The area under the ROC curve (AUC) is a measure of the model's ability to distinguish between classes. Hence, we further report the ROC curve which shows the trade-off between the true positive rate (TPR) and false positive rate (FPR) for each class for stance and dogmatism tasks, see Figs. 11 and. 12 in Appendix F.

**Verification using Human Interaction.** Due to the time-consuming nature of the manual annotation process, we perform human annotations on the set of 200 test conversations. In the forms for human annotations, we displayed the top 2 author's Reddit posts from the conversation, along with the submission title and content. We also provided a link to the original Reddit URL for annotators to look at the full conversation. We provided detailed annotation guidelines (similar to the ones mentioned in the prompt in Appendix B) to instruct human annotators in carrying out these tasks. Here is a sample Google form[6]. With three human annotators on a sample of 10 conversations, the agreement of majority labels (i.e., USDC test set labels) with human labels is 0.56 for the stance detection task and 0.45 for the dogmatism task. The annotators included two males and one female, affiliated with both academia and industry, aged between 20 and 40, and were very familiar with Reddit topics.

# 6 Conclusion

In this paper, we focused on the problems of 5-class stance and 4-class dogmatism classification in long conversations. Using LLMs as human-like annotators, we introduced USDC, a large-scale dataset of user stance and dogmatism in conversations. This is achieved by providing detailed annotation guidelines in the system prompt and full-length conversation as user prompt. Commercialized API-based LLMs generate author-level stance and dogmatism labels via zero, one and few-shot settings. The full-length multi-user conversation aspect of USDC allows it to capture the contextual and opinion shifts of multiple users in a conversation. We believe that the ability to perform finetuning or instruction tuning SLMs for user opinions at a large scale can bridge the gap between SLMs and commercial LLMs for understanding user traits. While finetuning SLMs shows F1-score on both stance and dogmatism tasks, the F1-score remains below 60% (54.9% for Stance and 51.4% for Dogmatism). On the other hand, instruction tuning of SLMs only improves F1-score performance on stance, not the dogmatism task. Further, the performance still falls short of 60%, with weighted F1-scores of 56.2% for stance and 49.2% for dogmatism. These findings indicate that there is still significant room for improvement in understanding user opinions from a text segment.

**Limitations.** We plan to extend this work along the following directions in the future. 1) We performed this work on English conversations only. It would be nice to extend this to multi-lingual conversations and verify how accurately SLMs and LLMs perform on the Stance and Dogmatism tasks in the multi-lingual scenario. 2) We analyzed user dogmatism based on their posts within a single conversation. This approach could be extended to include posts across multiple conversations and utilize similar profile information if available. 3) We analyzed dogmatism information for only the top two authors. This was mainly because considering more authors increases the output generation length, and we were constrained by our budget. This implies that our current models have not been evaluated for authors who do not post frequently.

---
[6]https://forms.gle/dbPQBsNyfNJjvUeR9

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

# A  Detailed Statistics of the USDC Dataset

Table 4 shows the detailed statistics of our USDC dataset at the sub-reddit level. Fig. 5 shows the distribution of stance labels across LLM annotations across zero-shot, one-shot and few-shot settings. Fig. 6 shows the distribution of dogmatism labels across LLM annotations across zero-shot, one-shot and few-shot settings.

Table 4: Statistics of the User Conversation Dataset.

| subreddit | num_conversations | min_total_token_count | max_total_token_count |
|---|---|---|---|
| DebateCommunism | 73 | 529 | 11557 |
| Abortiondebate | 70 | 1271 | 7401 |
| CapitalismVSocialism | 61 | 665 | 16927 |
| prochoice | 60 | 582 | 7278 |
| brexit | 56 | 637 | 4553 |
| climateskeptics | 56 | 734 | 7550 |
| prolife | 54 | 672 | 13342 |
| gunpolitics | 52 | 683 | 7889 |
| MensRights | 52 | 623 | 5774 |
| climatechange | 49 | 520 | 7427 |
| nuclear | 41 | 572 | 5282 |
| progun | 39 | 436 | 3632 |
| NuclearPower | 23 | 629 | 4589 |
| Vegetarianism | 22 | 627 | 3958 |
| AntiVegan | 20 | 351 | 5052 |
| climate | 13 | 701 | 4678 |
| Egalitarianism | 10 | 665 | 4060 |
| VeganActivism | 8 | 460 | 3685 |
| Veganism | 2 | 1332 | 1738 |
| AnimalRights | 1 | 845 | 845 |
| animalwelfare | 1 | 1363 | 1363 |
| GunsAreCool | 1 | 2945 | 2945 |

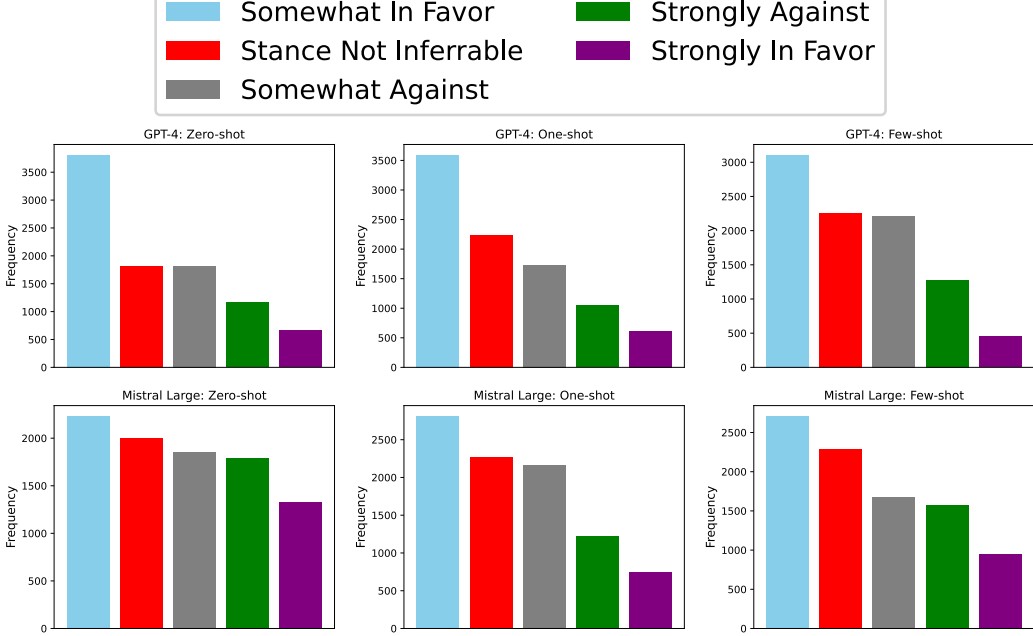

Figure 5: Distribution of Stance labels across LLM annotations.

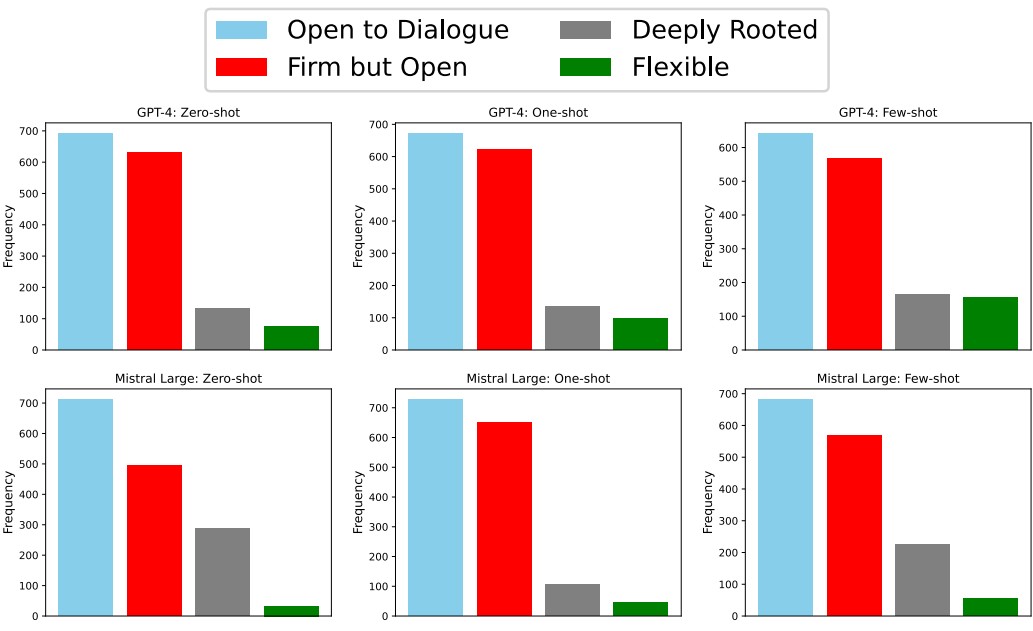

Figure 6: Distribution of dogmatism labels across LLM annotations.

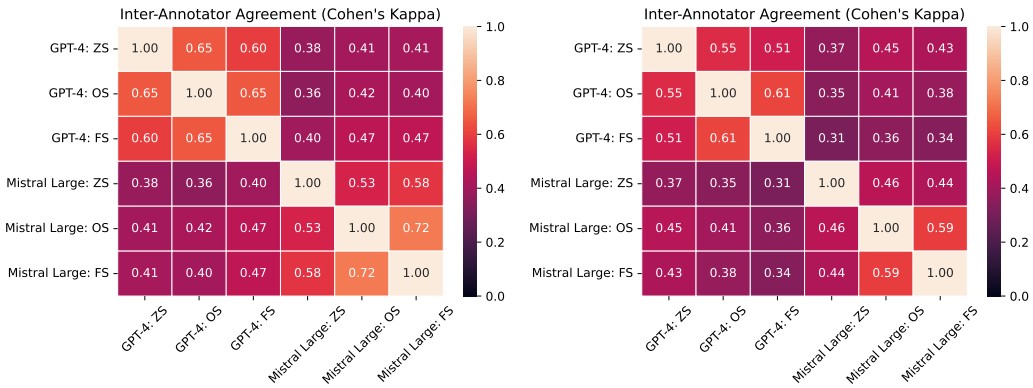

Figure 7: Inter-annotator agreement (IAA): Cohen's Kappa score across six different models (2 models×3 settings) for Stance (left) and Dogmatism (right) tasks.

## B System Prompt for LLM Annotation

We used the following prompt for getting annotations from LLMs as well as for instruction-tuning of SLMs.

```
"""
### Introduction
**Objective**: Analyze Reddit conversations to identify the stance of
    specific authors on sociopolitical topics and determine their level of
    dogmatism.
**Stance Definition**: Stance is defined as the expression of the author's
    standpoint and judgement towards a given topic.
**Dogmatism Definition**: Dogmatism is an opinion strongly believed as a fact
    to support a stance without a question or allowance for conversation.
**Task**: Given a JSON formatted Reddit submission and its comment thread,
    classify the stance of text segments related to ''author1'' and
    ''author2'' by assigning one of the following five predefined stance
    labels: 'strongly_against', 'somewhat_against', 'somewhat_in_favor',
    'strongly_in_favor', 'stance_not_inferrable'. Also, assign a dogmatism
```

```
790        label for each author by assigning one of the following four predefined
791        labels: 'Deeply Rooted', 'Firm but Open', 'Open to Dialogue', 'Flexible'.
792
793    ### Description of Stance Labels:
794    1. **strongly_against / strongly_in_favor**: Marks text showing strong
795       opinions, emotional expressions, or argumentative tones.
796    2. **somewhat_against / somewhat_in_favor**: Identifies texts with openness
797       to discussion, less certainty, or showing interest in different
798       viewpoints.
799    3. **stance_not_inferrable**: Use for texts that are neutral, support both
800       stances, or where the stance is unclear despite being on-topic.
801
802    ### Description of Dogmatism Labels:
803    1. **Deeply Rooted**: Reflects a strong, unchangeable belief. This label
804       conveys the idea of someone who is firm in their opinion and unlikely to
805       be swayed.
806    2. **Firm but Open**: Indicates a person who is not likely to change their
807       mind but does not impose their views authoritatively. It captures the
808       essence of being steadfast in one's beliefs without being dismissive of
809       others.
810    3. **Open to Dialogue**: Describes someone who holds a certain opinion but is
811       genuinely interested in considering other viewpoints. This label suggests
812       a willingness to engage in meaningful conversation about differing
813       perspectives.
814    4. **Flexible**: Denotes a person who is not firmly committed to their stance
815       and is open to changing their opinion. This label is indicative of
816       flexibility and openness to new information or arguments.
817
818    ### Input Data Format
819    The input data will be in JSON format and will include several key elements
820        to represent a Reddit submission and its associated comments. Each
821        element provides specific information as described below:
822
823    - 'id': This is the unique identifier for the Reddit submission.
824    - 'title': The title of the post. This is what users see first and often
825        summarizes or hints at the content of the submission.
826    - 'content': The main post's detailed description. This text segment provides
827        the core message or information the author wishes to communicate with the
828        Reddit community. It may include narratives, questions, or any
829        information relevant to the title.
830    - 'author1' or 'author2': The username of our focus author. This field is
831        applicable if the post or comment is made by one of the specific authors
832        we are tracking in the dataset.
833    - 'comments': An array (list) of comments related to the Reddit submission.
834        Each comment in this array includes the following fields:
835      - 'id': The unique identifier for the comment, allowing for identification
836          and reference within the dataset.
837      - 'author1' or 'author2': The username of the comment's author, if it is
838          made by one of our focus authors. This helps in tracking contributions
839          by specific individuals.
840      - 'body': The text of the comment. This is the main content of the comment
841          where the author responds to the post or another comment, providing
842          insights, opinions, or further information.
843      - 'replies': An array of comments that are direct responses to this
844          comment. The structure of each reply follows the same format as the
845          initial comment, including 'id', 'author1' or 'author2' (if
846          applicable), 'body', and potentially more 'replies'.
847
848    ### Output Data Format
849    Submit your annotations in JSON format, grouping all stance annotations under
850        the key ''stance_annotations''. Each entry should be a dictionary
851        containing the segment's ''id'', your ''label'', and the ''reason'' for
852        your choice. Include the dogmatism label and its justification under
853        ''dogmatism_label'' and ''dogmatism_reason'' keys, respectively.
854
```

```
855    The output should follow this structure:
856    ```json
857    {
858      "author1": {
859        "name": "[author_name]",
860        "stance_annotations": [
861          {
862            "id": "[segment_id]",
863            "label": "[chosen_label]",
864            "reason": "[Justification in <50 words]"
865          },
866          ...
867        ],
868        "dogmatism_label": "[chosen_dogmatism_label]",
869        "dogmatism_reason": "[Justification in <50 words]"
870      },
871      "author2": {
872        "name": "[author_name]",
873        "stance_annotations": [
874          {
875            "id": "[segment_id]",
876            "label": "[chosen_label]",
877            "reason": "[Justification in <50 words]"
878          },
879          ...
880        ],
881        "dogmatism_label": "[chosen_dogmatism_label]",
882        "dogmatism_reason": "[Justification in <50 words]"
883      }
884    }
885    '''
886    ### Instructions for Effective Annotation
887
888    1. **Labeling Stance**: For each segment (including the original Reddit
889       submission, comments, or replies) where "author1" or "author2" is
890       mentioned, assign a stance label that best represents the stance
891       expressed towards the discussed topic in the submission. This
892       comprehensive approach ensures no relevant contribution by "author1" or
893       "author2" is overlooked. Evaluate the stance based on the content's tone,
894       argumentation, and engagement level with the topic.
895    2. **Providing Justification**: For each label assigned, include a concise
896       reason, aiming for less than 50 words. Focus on the stance and
897       argumentative indicators present in the text.
898    3. **Dogmatism Assessment**: After reviewing all segments from "author1" and
899       "author2", assign a single dogmatism label reflecting the overall tone
900       and approach in their contributions.
901    """
```

## C  Prompts for Finetuning SLMs

Fig. 8 and 9 shows the prompts used for finetuning SLMs for the stance and dogmatism classification tasks respectively.

> **Stance Classification**
>
> Analyze the stance of the post enclosed in square brackets.
> Categorize each post into one of the following categories based on its stance:
>
> - Somewhat In Favor
> - Somewhat Against
> - Stance Not inferrable
> - Strongly In Favor
> - Strongly Against
>
> and return the answer as one of the corresponding stance labels.
>
> `[{data_point["stance_id_comment"]}]`

Figure 8: Prompt for stance classification, for finetuning SLMs.

> **User Dogmatism Identification**
>
> Analyze the comments of a user in conversation enclosed in square brackets.
> Categorize the opinion fluctuation of the user into one of the following categories based on its change:
>
> - Open to Dialogue
> - Firm but Open
> - Deeply Rooted
> - Flexible
>
> Return the answer as one of the corresponding dogmatism labels.
>
> `[{data_point["comments_string_for_dogmatism"]}]`

Figure 9: Prompt for dogmatism classification, for finetuning SLMs.

## D  Sample of User Input Prompt

```
"""
### User Prompt
Now complete the given task for the respective authors i.e., author1
    name is "rookerin0" and respective ids are ['dhoxyz', 'f3pghji', '
    f3tywb4', 'f3uomn2']. author2 name is "MikeWillTerminate" and
    respective ids are ['f3rt0bf', 'f3rqu2u'] for the data in json
    format
{
    "id":"dhoxyz",
    "author1":"rookerin0",
    "title":"This sub should encourage anti vs. pro-gun discussions
        instead of shutting them down instantly",
    "content":"Honesly, I followed this sub especifically to take part
        in these discussions, but everytime I see a comment that even
        remotely suggests anti gun ideals or a discussion on the
        subject just gets ignored and downvoted to hell. Kind of
        expecting this to go the same way (my karma anus is ready,
        downvotes) , but I have to hope for healthy discussions on the
        subject.",
    "comments":[
        {
            "id":"f3p9n2c",
            "body":"I think the problem now is the two sides are at an
                impasse. Everytime there is a "compromise" pro gun loses
```

```
931                     something. Now days pro gun is interpreting the
932                     Constitution more literal, which leaves even the most
933                     mild policies of anti gun as infringements. To further
934                     compound this anti gun is only considering the most
935                     extreme measures. "Assault Weapons" bans, mandatory
936                     buybacks, red flag laws, etc.. I think at this point
937                     there is just nothing left to talk about. The middle
938                     ground is gone.",
939               "replies":[
940                     {
941                         "id":"f3pati9",
942                         "replies":[
943                               {
944                                   "id":"f3pdu44",
945                                   "body":"You are exactly right. I'm done with the
946                                       idea that there can be real compromise. We
947                                       should have at least gotten national
948                                       reciprocity and shall-issue in every state in
949                                       exchange for what we've given up. Now you
950                                       have to be a goddamn lawyer to exercise your
951                                       rights without violating the law."
952                               },
953                               {
954                                   "author2":"MikeWillTerminate",
955                                   "id":"f3rt0bf",
956                                   "body":"I am prepared for UBCs, if they do this:
957                                   1. Lower the age to buy handguns to 18,
958                                       nationwide.
959                                   2. Repeal the Hughes Amendment:
960                                   3. A FOPA-like ban on assault weapon bans (what
961                                       the FOPA did with a registry)
962                                   4. The punishment for violation is a monetary
963                                       fine only
964                                   5. A repeal of the GCA ban on foreign NFA weapons
965                                   6. A repeal of the National Minimum Drinking Age
966                                       Act of 1984"
967                               }
968                         ]
969                     },
970                     {
971                         "id":"f3pd55z",
972                         "body":"Everytime there is a "compromise" pro gun loses
973                             something. That and today's compromise is tomorrow
974                             's loophole to be closed. All such compromises do
975                             is push that policy off until the next round."
976                     }
977               ]
978         },
979         {
980             "id":"f3paf0j",
981             "body":"Yeah this sub it's not conducive to conversion. Its
982                 quickly devolving to little more than "Boogaloo" memes
983                 and shouting "SHALL. NOT." at each other. However, as
984                 far as I know, the mods won't delete your thread and ban
985                 you from the sub for trying to have a good faith
986                 discussion, like some of the gun control subs will.",
987             "replies":[
988                   {
989                       "id":"f3pusbm",
990                       "body":"Unfortunately this sub's mod team takes a very
991                           passive approach to moderation. With very little
992                           effort they could make this sub into a quality
993                           progun meeting ground *without having to resort to
994                           censorship*. Instead they promote low-effort memes
995                           and endless duplication of posts through their
```

```json
                        inaction. whubbard has the chops to resurrect this
                        sub. Let's see if he's up to the challenge.",
                "replies":[
                    {
                        "id":"f3q8xj6",
                        "body":"We voted to ban memes last week. All
                            about rolling it out now.",
                        "replies":[
                            {
                                "id":"f3qn4p8",
                                "body":"Damn I might have to eat some crow
                                    here then..."
                            }
                        ]
                    }
                ]
            }
        ]
    },
    {
        "id":"f3pafqa",
        "body":"Found the gun grabber!!",
        "replies":[
            {
                "id":"f3pcw4h",
                "body":"Witch hunter."
            }
        ]
    },
    {
        "id":"f3pal5l",
        "body":"I see people have discussions when it makes sense to.
            Not much reason to spend time responding to the same gun
            control measures over and over though."
    },
    {
        "id":"f3paw3h",
        "body":"I get where you're coming from, but people's ability
            to protect themselves and own their own property isn't
            something that is compromisable. Anything less, and they
            cease to own their own property. It's like breathing,
            there can be nothing less than total ability to breath
            when and how someone wants. It's just that simple."
    },
    {
        "id":"f3pax9m",
        "body":"My take on this, What kind of open discussion is
            possible for a right that is guaranteed and most
            importantly, not to be infringed upon? They're making all
             these unlawful laws to portray it as it's somehow
            legitimate. They are not, We are at an apex, to which
            both political spectrums and even us to a degree are
            liable for.\nI certainly believe both sides are waiting
            for this to boil over so each can finger point. I just
            speculate it's going to be the hell humanity been
            whispering about but never thought it would ever occur."
    },
    {
        "id":"f3pb6ny",
        "body":"The time for discussion is over."
    },
    {
        "id":"f3pfqwq",
        "body":"I don't know what you're talking about. Sure people
            downvote, but they also talk. We get "why do you need
```

```
1061                    guns" posts at least weekly, and several people will
1062                    engage in actual conversation with them, citing facts,
1063                    clearing up statistics, and telling stories to illustrate
1064                     why this is important to them, but they are usually met
1065                    with "you stupid @#$%, you think you're Rambo" or
1066                    something equally clever. People who come here to discuss
1067                     and learn will be treated well. People who are just
1068                    trolling are treated like trolls.",
1069                "replies":[
1070                    {
1071                        "author1":"rookerin0",
1072                        "id":"f3pghji",
1073                        "body":"I made this post because I'm always seeing
1074                            rational, conversation seeking comments getting
1075                            blown to downvote hell.",
1076                        "replies":[
1077                          {
1078                          "id":"f3pi9xv",
1079                          "body":"[Like this one?](https://www.reddit.com/r/
1080                              progun/comments/dhcu92/yup/f3p75tg/)> One smart
1081                              man in a sub full of... welp... "strong opinions
1082                              ". You start off with arrogance, as the sole
1083                              arbiter of what constitutes a "smart man". Then
1084                              you back it up with a dismissive swipe at what
1085                              you term "strong opinions".> Every other country
1086                              can see that PROPER gun control reduces gun
1087                              violence by a ton, More arrogance. False
1088                              equivalence. Unsupported claims.> but the US
1089                              refuses to let go of it's antique laws In a
1090                              shocking turn of events, more arrogance.> Fully
1091                              aware that this is a fully pro gun sub, willing
1092                              to take the downvotes in order to spark a
1093                              discussion and crack some heads. You aren't the
1094                              first arrogant asshole to grace this sub with
1095                              posts like this. Try bringing something other
1096                              than your own self-importance to the discussion.
1097                              Edit: And then there's [this gem](https://www.
1098                              reddit.com/r/unpopularopinion/comments/d3w5z1/
1099                              people_living_in_the_us_are_living_in_one_of_the/
1100                              f06r3sg/.> Wanna feel like you could be shot at
1101                              every single moment? Move to the US, it'll prob
1102                              happen to you either as a bystander, or you'd be
1103                              shot by a random citizen (sometimes police)."
1104                          },
1105                          {
1106                              "id":"f3pj8k0",
1107                              "body":"As is tradition. We're done with that
1108                                  condescending bullshit from antis, you dont
1109                                  come here for good faith discussion and
1110                                  whether you get a reasonable response or not,
1111                                   nothing ever changes, easier to downvote you
1112                                   and move on because we get the same
1113                                  treatment anytime we attempt to speak out in
1114                                  anti subs."
1115                          },
1116                          {
1117                              "id":"f3plgf4",
1118                              "body":"If downvotes hurt your feelings, you
1119                                  shouldn't be on reddit. People tend to
1120                                  downvote anything they disagree with (which
1121                                  is why some subs specifically ask you to only
1122                                   downvote things that contribute nothing to
1123                                  the discussion). It's a bad habit, but that's
1124                                   the way it is. People downvote and *still*
1125                                  enage. You want to post a view contrary to
```

```
1126                           the prevailing view of the sub, take your
1127                           lumps and participate in what conversation
1128                           you are offered. But if you're only here to
1129                           preach about how stupid, misguided, unevolved
1130                           , uneducated, irrational, and/or violent we
1131                           are, don't expect a polite response."
1132                      },
1133                      {
1134                          "id":"f3tcgf1",
1135                          "body":"An arrogant Israeli trying to tell
1136                              another nation how they should be run. You're
1137                               just a walking stereotype aren't you? And
1138                              before you say anything, I popped into your
1139                              comment history. That's where the calling you
1140                               Israeli comes from.",
1141                          "replies":[
1142                              {
1143                                  "author1":"rookerin0",
1144                                  "id":"f3tywb4",
1145                                  "body":"I thought that trying to tell other
1146                                       nations how they should run was your
1147                                      guys's stereotype.",
1148                                  "replies":[
1149                                      {
1150                                          "id":"f3u0vkq",
1151                                          "body":"No we go in and try to make
1152                                              them work our way."
1153                                      }
1154                                  ]
1155                              }
1156                          ]
1157                      }
1158                  ]
1159              }
1160          ]
1161      },
1162      {
1163          "id":"f3pzseh",
1164          "body":"It's a little unfortunate but the grabbers who come
1165              on here tend to be intellectually dishonest and/or
1166              uninformed. There was some Australian post a few days ago
1167               that pretty much asked why we like our guns more than
1168              children. No discussion to be had there. There's also
1169              some posts that clearly demonstrate the poster should
1170              inform himself or herself a little."
1171      },
1172      {
1173          "author2":"MikeWillTerminate",
1174          "id":"f3rqu2u",
1175          "body":"Actually, do that. It shows everyone that they tend
1176              to be crazy, unstable, ignorant, stereotyping, arrogant
1177              bastards who hate black people with a hair trigger."
1178      },
1179      {
1180          "id":"f3t7tgg",
1181          "body":"Welcome to reddit, home of every single safe place
1182              for anything that doesnt violate the TOS. At least its
1183              slightly better than r/politics"
1184      },
1185      {
1186          "id":"f3unt9z",
1187          "body":"This isn't r/gundebate. This is a pro gun subreddit.
1188               That said, we do allow some debate provided it remains
1189              civil.",
1190          "replies":[
```

```
1191              {
1192                  "author1":"rookerin0",
1193                  "id":"f3uomn2",
1194                  "body":"Sadly tho, r/gundebate is pretty dead..."
1195              }
1196          ]
1197      },
1198      {
1199          "id":"f4dip6o",
1200          "body":"Anything else you want to give away for free?"
1201      }
1202      ]
1203  }
1204
1205  #### Response
1206
```

# E  Samples of JSON Outputs from LLMs

### E.1  Sample JSON Output from LLM: GPT-4 Zero-shot

```
1209
1210
1211  {
1212      "author1": {
1213      "name": "rookerin0",
1214      "stance_annotations": [
1215      {
1216          "id": "dhoxyz",
1217          "label": "somewhat_in_favor",
1218          "reason": "Advocates for open discussions, showing interest in
1219              dialogue."
1220      },
1221      {
1222          "id": "f3pghji",
1223          "label": "somewhat_in_favor",
1224          "reason": "Expresses desire for rational discussions, despite
1225              downvotes."
1226      },
1227      {
1228          "id": "f3tywb4",
1229          "label": "somewhat_against",
1230          "reason": "Sarcastic retort, but lacks strong emotional
1231              language."
1232      },
1233      {
1234          "id": "f3uomn2",
1235          "label": "stance_not_inferrable",
1236          "reason": "Comment is factual, no clear stance on the debate
1237              itself."
1238      }
1239      ],
1240      "dogmatism_label": "Open to Dialogue",
1241      "dogmatism_reason": "Shows interest in discussion, acknowledges
1242          downvotes."
1243      },
1244      "author2": {
1245      "name": "MikeWillTerminate",
1246      "stance_annotations": [
1247      {
1248          "id": "f3rt0bf",
1249          "label": "somewhat_in_favor",
1250          "reason": "Proposes conditions for UBCs, indicating some
1251              openness."
1252      },
```

```
1253    {
1254        "id": "f3rqu2u",
1255        "label": "strongly_against",
1256        "reason": "Uses strong negative descriptors for opponents,
1257            indicating firm stance."
1258    }
1259    ],
1260    "dogmatism_label": "Firm but Open",
1261    "dogmatism_reason": "Mixed; proposes solutions but also dismisses
1262        opposition."
1263    }
1264 }
1265
```

## E.2  Sample JSON Output from LLM: GPT-4 One-shot

```
1267
1268 {
1269     "author1": {
1270     "name": "rookerin0",
1271     "stance_annotations": [
1272     {
1273         "id": "dhoxyz",
1274         "label": "somewhat_in_favor",
1275         "reason": "Advocates for open discussions, indicating a pro-
1276             discussion stance."
1277     },
1278     {
1279         "id": "f3pghji",
1280         "label": "somewhat_in_favor",
1281         "reason": "Expresses desire for rational conversation, showing
1282             a supportive stance for dialogue."
1283     },
1284     {
1285         "id": "f3tywb4",
1286         "label": "somewhat_against",
1287         "reason": "Uses sarcasm to counter a stereotype, indicating a
1288             stance against the opposing view."
1289     },
1290     {
1291         "id": "f3uomn2",
1292         "label": "somewhat_in_favor",
1293         "reason": "Expresses disappointment in lack of active debate,
1294             supporting open discussion."
1295     }
1296     ],
1297     "dogmatism_label": "Open to Dialogue",
1298     "dogmatism_reason": "Shows a consistent preference for discussion
1299         and engagement, without dogmatic assertions."
1300     },
1301     "author2": {
1302     "name": "MikeWillTerminate",
1303     "stance_annotations": [
1304     {
1305         "id": "f3rt0bf",
1306         "label": "somewhat_in_favor",
1307         "reason": "Lists conditions for agreement, showing openness to
1308             compromise."
1309     },
1310     {
1311         "id": "f3rqu2u",
1312         "label": "strongly_against",
1313         "reason": "Uses strong language to describe opponents,
1314             indicating a firm stance against them."
1315     }
```

```
1316        ],
1317            "dogmatism_label": "Firm but Open",
1318            "dogmatism_reason": "While showing strong opinions, also lists
1319                specific conditions for compromise."
1320        }
1321 }
1322
```

## E.3    Sample JSON Output from LLM: GPT-4 large Few-shot

```
1324
1325 {
1326     "author1": {
1327     "name": "rookerin0",
1328     "stance_annotations": [
1329     {
1330         "id": "dhoxyz",
1331         "label": "somewhat_in_favor",
1332         "reason": "Expresses desire for open discussions, indicating a
1333             stance in favor of dialogue."
1334     },
1335     {
1336         "id": "f3pghji",
1337         "label": "somewhat_in_favor",
1338         "reason": "Advocates for rational conversation despite
1339             observing negative reactions."
1340     },
1341     {
1342         "id": "f3tywb4",
1343         "label": "somewhat_against",
1344         "reason": "Uses sarcasm to counter a stereotype, showing a
1345             stance against it."
1346     },
1347     {
1348         "id": "f3uomn2",
1349         "label": "stance_not_inferrable",
1350         "reason": "Comment on the state of another subreddit does not
1351             reveal a stance on the gun debate."
1352     }
1353     ],
1354     "dogmatism_label":  "Open to Dialogue",
1355     "dogmatism_reason": "Shows a consistent
1356     preference for discussion over confrontation."
1357     },
1358     "author2": {
1359     "name": 'MikeWillTerminate',
1360     "stance_annotations": [
1361     {
1362         "id": 'f3rt0bf',
1363         "labe"': 'somewhat_in_favor",
1364         "reason": 'Lists conditions for accepting UBCs, indicating
1365             some openness to compromise."
1366     },
1367     {
1368         "id": 'f3rqu2u',
1369         "label": "strongly_against",
1370         "reason": "Uses derogatory language to describe opponents,
1371             showing strong opposition."
1372     }
1373     ],
1374     "dogmatism_label": "Firm but Open",
1375     "dogmatism_reason": "While showing strong opinions, also suggests
1376         conditions for policy acceptance."
1377     }
1378 }
1379
```

## E.4 Sample JSON Output from LLM: Mistral Large Zero-shot

```
{
    "author1": {
    "name": "rookerin0",
    "stance_annotations": [
    {
        "id": "dhoxyz",
        "label": "somewhat_in_favor",
        "reason": "Author expresses desire for open discussions on
            anti-gun ideals."
    },
    {
        "id": "f3pghji",
        "label": "stance_not_inferrable",
        "reason": "Author is commenting on downvotes, not expressing
            stance."
    },
    {
        "id": "f3tywb4",
        "label": "stance_not_inferrable",
        "reason": "Author is making a stereotypical comment, not
            expressing stance."
    },
    {
        "id": "f3uomn2",
        "label": "stance_not_inferrable",
        "reason": "Author is commenting on the state of another
            subreddit, not expressing stance."
    }
    ],
    "dogmatism_label": "Open to Dialogue",
    "dogmatism_reason": "Author expresses desire for open discussions
        and does not dismiss opposing views."
    },
    "author2": {
    "name": "MikeWillTerminate",
    "stance_annotations": [
    {
        "id": "f3rt0bf",
        "label": "strongly_in_favor",
        "reason": "Author lists conditions for accepting UBCs, showing
            strong pro-gun stance."
    },
    {
        "id": "f3rqu2u",
        "label": "strongly_in_favor",
        "reason": "Author"s comment is dismissive of anti-gun
            advocates, showing strong pro-gun stance."
    }
    ],
    "dogmatism_label": "Deeply Rooted",
    "dogmatism_reason": "Author"s comments are dismissive of opposing
        views and show a strong, unchangeable belief in pro-gun ideals
        ."
    }
}
```

## E.5 Sample JSON Output from LLM: Mistral Large One-shot

```
{
    "author1": {
    "name": "rookerin0",
```

```
1443        "stance_annotations": [
1444        {
1445            "id": "dhoxyz",
1446            "label": "somewhat_in_favor",
1447            "reason": "Expresses desire for open discussions on gun
1448                control, but follows pro-gun subreddit."
1449        },
1450        {
1451            "id": "f3pghji",
1452            "label": "stance_not_inferrable",
1453            "reason": "Comments on downvotes, but no clear stance on gun
1454                control."
1455        },
1456        {
1457            "id": "f3tywb4",
1458            "label": "stance_not_inferrable",
1459            "reason": "Comment about stereotypes, but no clear stance on
1460                gun control."
1461        },
1462        {
1463            "id": "f3uomn2",
1464            "label": "stance_not_inferrable",
1465            "reason": "Mentions r/gundebate being dead, but no clear
1466                stance on gun control."
1467        }
1468        ],
1469        "dogmatism_label": "Open to Dialogue",
1470        "dogmatism_reason": "Seeks open discussions and does not express
1471            strong dogmatic views."
1472        },
1473        "author2": {
1474        "name": "MikeWillTerminate",
1475        "stance_annotations": [
1476        {
1477            "id": "f3rt0bf",
1478            "label": "somewhat_in_favor",
1479            "reason": "Lists conditions for supporting UBCs, showing
1480                openness to compromise."
1481        },
1482        {
1483            "id": "f3rqu2u",
1484            "label": "strongly_in_favor",
1485            "reason": "Strong negative language against gun control
1486                advocates, suggesting strong pro-gun stance."
1487        }
1488        ],
1489        "dogmatism_label": "Firm but Open",
1490        "dogmatism_reason": "Expresses strong pro-gun views but open to
1491            compromise under certain conditions."
1492        }
1493 }
1494
```

## E.6   Sample JSON Output from LLM: Mistral Large Few-shot

```
1496
1497 {
1498        "author1": {
1499        "name": "rookerin0",
1500        "stance_annotations": [
1501        {
1502            "id'": "dhoxyz",
1503            "label": "somewhat_in_favor",
1504            "reason": "Expresses desire for open discussions on anti vs.
1505                pro-gun debates."
```

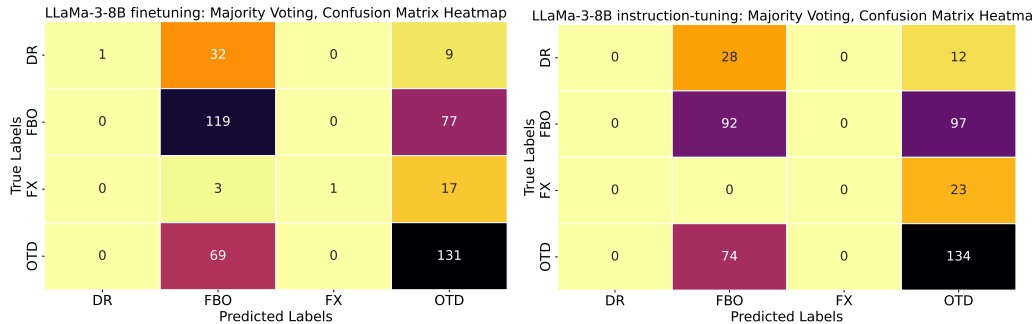

Figure 10: Confusion matrix for LLaMa-3-8B Dogmatism detection models on USDC test set: finetuning (left) and instruction-tuning (right). Here, DR: Deeply Rooted, FX: Flexible, FBO: Firm but Open, OTD: Open to Dialogue

```
1506        },
1507        {
1508            "id": "f3pghji",
1509            "label": "stance_not_inferrable",
1510            "reason": "Comments on downvotes, but no clear stance on the
1511                topic."
1512        },
1513        {
1514            "id": "f3tywb4",
1515            "label": "stance_not_inferrable",
1516            "reason": "Makes a stereotypical comment, but no clear stance
1517                on the topic."
1518        },
1519        {
1520            "id": "f3uomn2",
1521            "label": "stance_not_inferrable",
1522            "reason": "Mentions r/gundebate being dead, but no clear
1523                stance on the topic."
1524        }
1525        ],
1526        "dogmatism_label": "Open to Dialogue",
1527        "dogmatism_reason": "Seeks open discussions and engages in
1528            conversation without strong dogmatic language."
1529        },
1530        "author2": {
1531        "name": "MikeWillTerminate",
1532        "stance_annotations": [
1533        {
1534            "id": "f3rt0bf",
1535            "label": "somewhat_in_favor",
1536            "reason": "Lists conditions for supporting UBCs, showing
1537                openness to discussion."
1538        },
1539        {
1540            "id": "f3rqu2u",
1541            "label": "strongly_against",
1542            "reason": "Uses derogatory language to express strong
1543                opposition to 'grabbers'."
1544        }
1545        ],
1546        "dogmatism_label": "Firm but Open",
1547        "dogmatism_reason": "Expresses strong opinions but also shows
1548            willingness to consider certain conditions for compromise."
1549        }
1550    }
1551
```

## F SLM finetuning: AUC (Area Under the Curve) analysis

Fig. 10 illustrates the confusion matrix for dogmatism detection for LLaMa-3-8B finetuning and instruction-tuning. We make the following observations from Fig. 10: 1) For both finetuning and instruction-tuning, there is significant misclassifications, especially for the "Deeply Rooted" and "Flexible" labels, with both having zero accuracy and F1-scores. While "Firm but Open" and "Open to Dialogue" perform moderately better, with accuracies of 48.7% and 64.4% respectively. The confusion matrix indicates substantial confusion to distinguish between intermediate levels of dogmatism, such as "Firm but Open" and "Open to Dialogue. We further reports the ROC curve shows the trade-off between the true positive rate (TPR) and false positive rate (FPR) for each class for stance and dogmatism tasks, in Figs. 11 and. 12. The area under the ROC curve (AUC) is a measure of the model's ability to distinguish between classes.

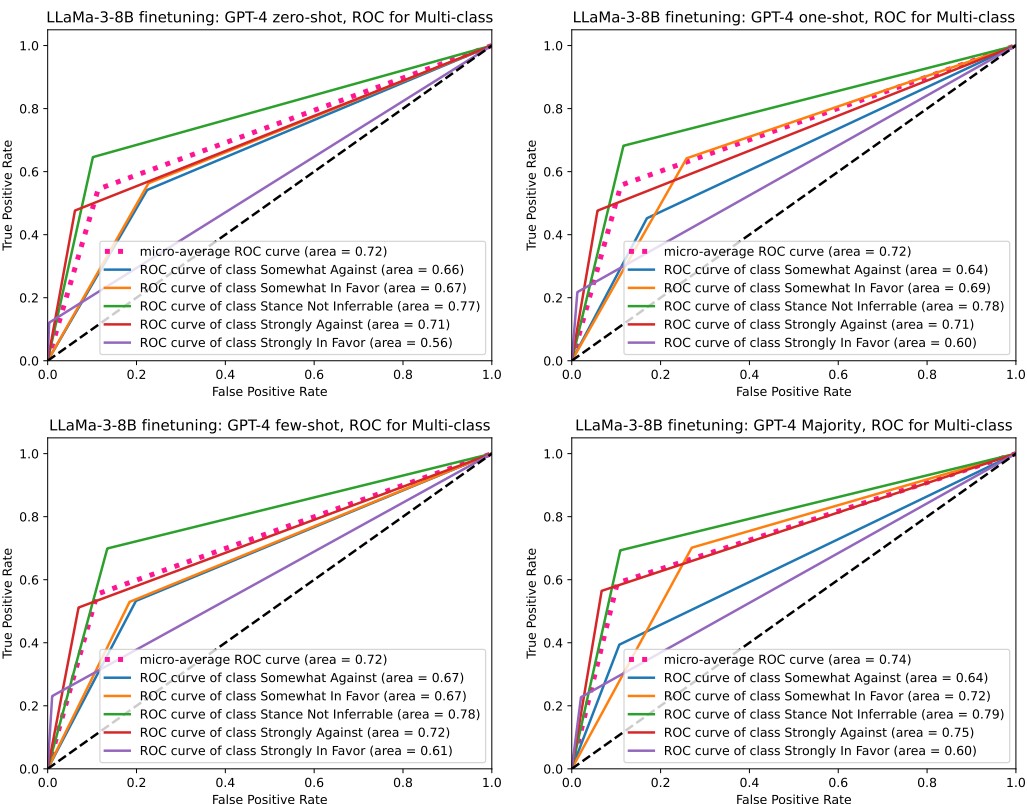

Figure 11: LLaMa-3-8B finetuning for stance detection task: Visualize the ROC curves for each class along with their AUC values for GPT-4 Annotations across zero-shot, one-shot, few-shot and majority labels.

## G SLM instruction-tuning: AUC (Area Under the Curve) analysis

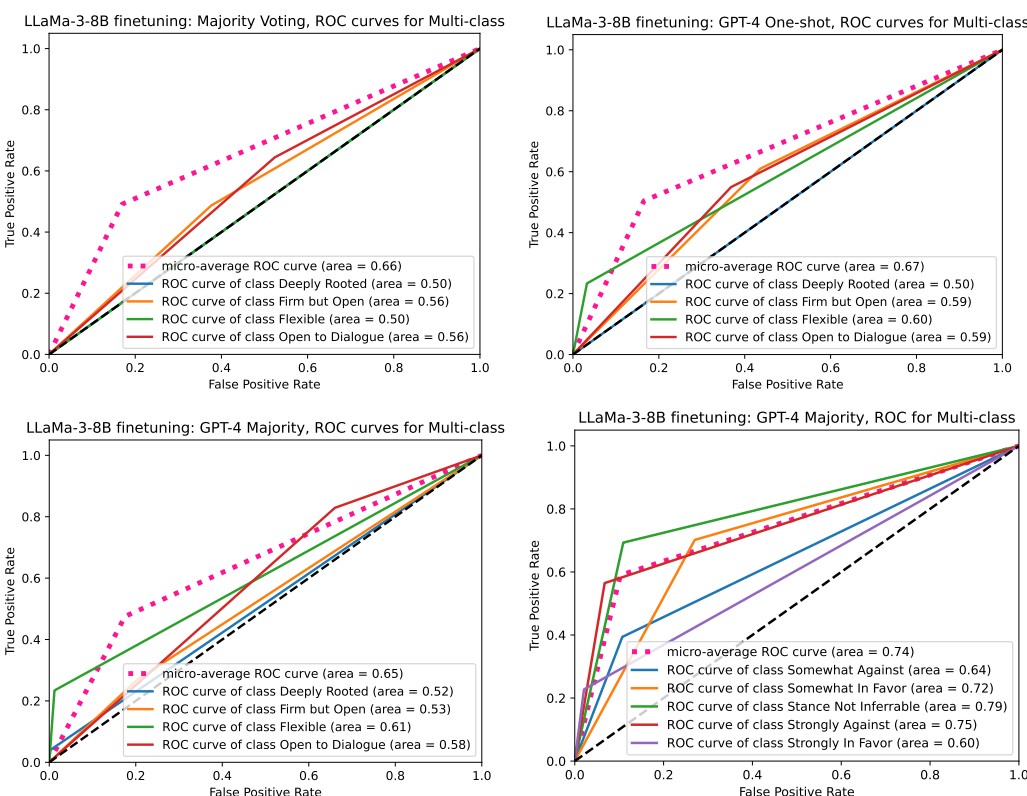

Figure 12: LLaMa-3-8B finetuning for dogmatism task: Visualize the ROC curves for each class along with their AUC values for GPT-4 Annotations across zero-shot, one-shot, few-shot and majority labels.

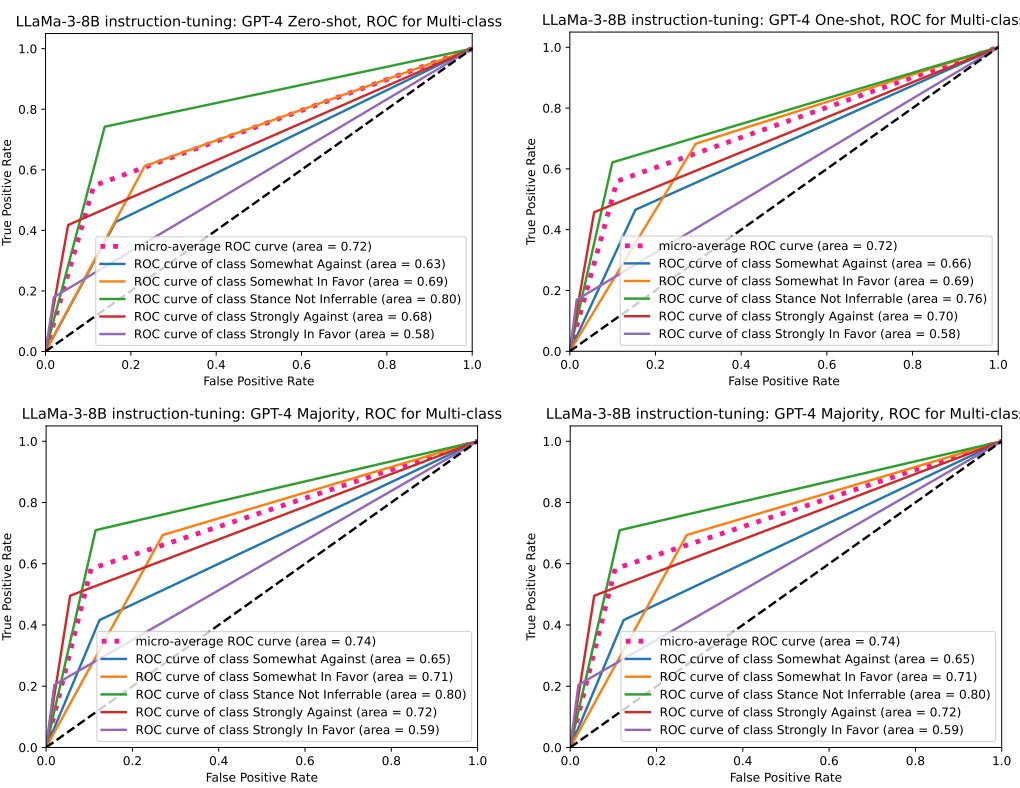

Figure 13: LLaMa-3-8B instruction-tuning for stance detection task: Visualize the ROC curves for each class along with their AUC values for GPT-4 Annotations across zero-shot, one-shot, few-shot and majority labels.

