# USDC: A Dataset of User Stance and Dogmatism in Long Conversations

## 1 Stance detection: Cross-modal comparison

To evaluate the LLM generated annotations, we now perform transfer learning by fine-tuning the SLMs on the USDC dataset. We then test the model's performance on the SPINOS dataset for a 5-class Stance detection task, as described by Sakketou et al. (2022). We use the same training dataset mentioned in Section 4.2 of the main paper. For testing, we use the SPINOS dataset, which consists of 3,238 post level examples across five Stance labels.

Fig. 1 illustrates the confusion matrix for Stance detection for LLaMa-3-8B finetuning on USDC and transfer learning on SPINOS. We make the following observations the Fig. 1: 1) There is a significant misclassification across all classes, with the "Stance Not Inferrable" label being the most commonly predicted class, resulting in many false positives for this label. 2) The model performs best in terms of accuracy for three stance classes: "Somewhat In Favor" (0.456), "Strongly Against" (0.400), and "Somewhat Against" (0.381), while performing the worst for the "Strongly In Favor" Stance (0.115). These overlaps suggest challenges in distinguishing whether a post contains Stance or not, indicating a need for enhanced feature representation and clearer class definitions to improve model performance.

In comparison to the SPINOS dataset results reported in the paper by Sakketou et al. (2022), where the best model achieved an F1-score of 0.341, a random baseline achieved 0.230, and a majority baseline achieved 0.124, our approach using LLaMa-3-8B finetuning on the USDC dataset achieved a weighted F1-score of 0.320 on SPINOS. This score is close to the best model performance on the SPINOS dataset, indicating that our LLM-generated annotations on the USDC dataset reach human-level performance on the SPINOS dataset. It is important to note that our weighted F1-score is significantly impacted by the "Stance Not Inferrable" class, which comprises the majority of samples in the SPINOS dataset. Our fine-tuned SLM struggled to classify this class accurately, leading to a lower overall weighted F1-score.

We also validated the SPINOS performance using other SLMs such as LLaMa-3-8B-Instruct, LLaMa-2-7B, LLaMa-2-7B-Chat and Vicuna-7B models. Figs. 2, 3, 4 and 5 display these model results. Observations from these figures indicate that these models report weighted F1-scores of 0.32, 0.305, 0.286, and 0.291. These results show that all models perform better than the random and majority baselines. Additionally, the LLaMa-3-8B-Instruct model's performance is close to the SPINOS benchmark on the 5-class Stance detection task.

In conclusion, the results indicate that LLM-generated annotations of USDC dataset are a viable alternative to human labels for Stance detection tasks, demonstrating substantial potential for automating and scaling up such complex annotation processes in long user conversation data.

## References

Flora Sakketou, Allison Lahnala, Liane Vogel, and Lucie Flek. Investigating user radicalization: A novel dataset for identifying fine-grained temporal shifts in opinion. *arXiv preprint arXiv:2204.10190*, 2022.

Figure 1: Confusion matrix for LLaMa-3-8B Stance detection models on SPINOS test set: finetuning on USDC and test it on SPINOS. SOA: Somewhat Against, SOIF: Somewhat In Favor, SNI: Stance Not Inferrable, SGA: Strongly Against, SIF: Strongly In Favor.

Figure 2: Confusion matrix for LLaMa-3-8B-instruct Stance detection models on SPINOS test set: finetuning on USDC and test it on SPINOS. SOA: Somewhat Against, SOIF: Somewhat In Favor, SNI: Stance Not Inferrable, SGA: Strongly Against, SIF: Strongly In Favor.

Figure 3: Confusion matrix for LLaMa-2-7B Stance detection models on SPINOS test set: finetuning on USDC and test it on SPINOS. SOA: Somewhat Against, SOIF: Somewhat In Favor, SNI: Stance Not Inferrable, SGA: Strongly Against, SIF: Strongly In Favor.

Figure 4: Confusion matrix for LLaMa-2-7B-chat Stance detection models on SPINOS test set: finetuning on USDC and test it on SPINOS. SOA: Somewhat Against, SOIF: Somewhat In Favor, SNI: Stance Not Inferrable, SGA: Strongly Against, SIF: Strongly In Favor.

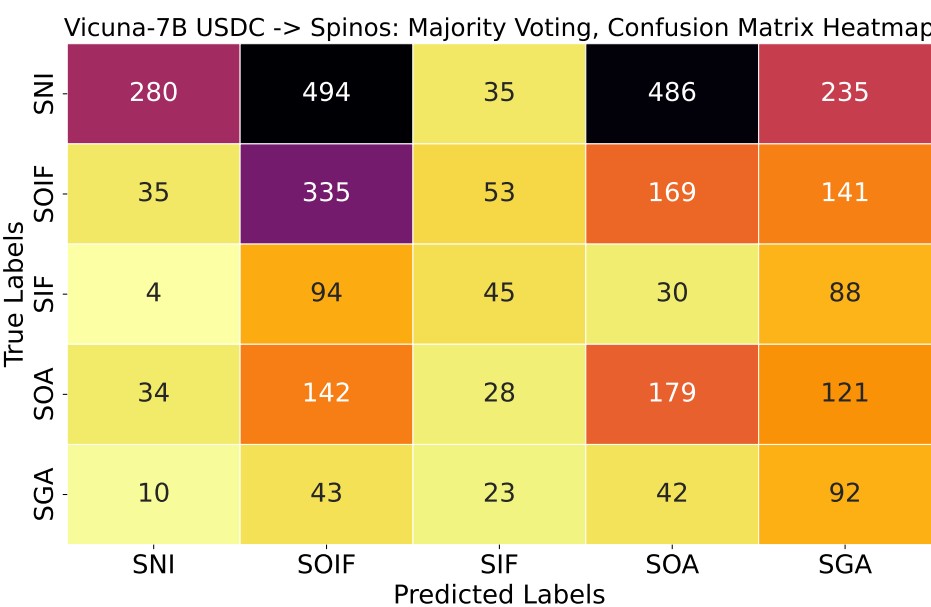

Figure 5: Confusion matrix for Vicuna-7B Stance detection models on SPINOS test set: finetuning on USDC and test it on SPINOS. SOA: Somewhat Against, SOIF: Somewhat In Favor, SNI: Stance Not Inferrable, SGA: Strongly Against, SIF: Strongly In Favor.