# OpenReview forum: "USDC: A Dataset of $\underline{U}$ser $\underline{S}$tance and $\underline{D}$ogmatism in Long $\underline{C}$onversations"
_NeurIPS.cc/2024/Datasets_and_Benchmarks_Track — Submitted to NeurIPS 2024 Track Datasets and Benchmarks_

### Official Review · Reviewer_DbAk · 2024-07-24
**Cool dataset, but what about the labels?**

**Rating:** 6
**Confidence:** 4
**Correctness:** Yes.
**Clarity:** Yes.

**Review:**

The paper is very thorough and well-written, with the authors explaining their inclusion and exclusion decisions and some of the complications they faced. The dataset represents a realistic use case  and is large enough testing and fine-tuning. The fact that the labels appear strictly to come from LLMs is a weakness.

**Strengths:**

+Well-written
+Informative evaluation
+Useful dataset (at least the inputs)

**Additional Feedback:**

N/A

**Documentation:**

Yes.

**Ethics:**

No.

**Limitations:**

See above discussion about lack of human labels.

**Opportunities For Improvement:**

I feel that, relying only on LLMs to label the data makes the labels unreliable. The authors of the paper could have at least labeled some of the data themselves and the report agreement levels with themselves and the LLMs.
Also, the work seems strongly related to sentiment analysis I would like (though this is not critical) a discusion of how this work relates to it.

**Relation To Prior Work:**

See discussion about about sentiment analysis.

**Summary And Contributions:**

The authors contribute a dataset of 764 reddit conversations, where 9,618 posts are labeled by stance toward the main issue raised in each top-level post and 1528 users are labeled according to their dogmatic orientation. All labels come from LLMs.

---

> ### Author Rebuttal · Authors · 2024-08-16
>
> *We thank the reviewer for their insightful and valuable comments and suggestions which are crucial for further strengthening our manuscript.*
>
> **Q1. The authors of the paper could have at least labeled some of the data themselves and the report agreement levels with themselves and the LLMs.**
>
> Thank you for raising this concern.
> * We would like to clarify that we have completed human annotation on 200 test conversations. We employed three human annotators to label these 200 test conversations using Google Forms (with 10 samples of conversations per form). The inter-annotator agreement between the majority labels (i.e., USDC test set labels) and the human labels was 0.49 for the stance detection task and 0.40 for the dogmatism task.
> * We have reported the IAA for both humans and LLMs for the stance and dogmatism tasks in CQ3 under the common responses. Kindly refer to common responses for detailed information.
>
> * Following the reviewer's suggestion, we also calculated the inter-annotator agreement among the three human annotators themselves, obtaining an agreement of 0.61 for the stance detection task and 0.52 for the dogmatism task.
> * We also addressed how USDC differs from prior studies and discussed its transfer learning performance on other existing datasets in our responses to CQ1 and CQ2. Kindly refer to those common questions for detailed information.
>
>
> **Q2. Also, the work seems strongly related to sentiment analysis I would like (though this is not critical) a discussion of how this work relates to it.**
>
> Thank you for this question.
> * We recognize that there is a conceptual overlap, as both fields involve the categorization and analysis of subjective content in text.
> * However, our work focuses on more specific tasks, such as stance detection and dogmatism identification, which go beyond the scope of traditional sentiment analysis. While sentiment analysis primarily aims to classify text into positive, negative, or neutral categories, our approach is concerned with understanding a user’s stance on a particular topic or the degree of dogmatism expressed in their statements.
> * These tasks require a different set of features and a more nuanced interpretation of language, which is not typically the focus of sentiment analysis. That being said, there are shared methodologies between our work and sentiment analysis, while our work is distinct in its goals and applications.

---

> > ### Comment · Reviewer_DbAk · 2024-08-28
> > **Thank you**
> >
> > Thank you for your rebuttal. I have decided to raise my score.

---

> > > ### Author Response · Authors · 2024-08-28
> > >
> > > We appreciate the reviewer's feedback and are confident that it has enhanced the paper's quality.

---

### Official Review · Reviewer_w9xr · 2024-07-25

**Rating:** 4
**Confidence:** 4

**Review:**

Pros:

- The writing of this paper is generally clear.

- The proposed dataset, particularly its focus on stance changes and dogmatism in long conversations, is a valuable contribution.

Cons:

- The paper proposed the conversational stance-annotated dataset but does not discuss in detail its advantages compared to existing conversational stance-related datasets. This makes it difficult for readers to assess the contribution and originality of the paper. I list a few conversational stance-related datasets below and suggest that the authors provide a thorough comparison and analysis with these datasets.

   [1] A Challenge Dataset and Effective Models for Conversational Stance Detection

   [2] Stance in replies and quotes (srq): A new dataset for learning stance in Twitter conversations

   [3] Improved target-specific stance detection on social media platforms by delving into conversation threads

- The annotation process of the proposed dataset relies entirely on LLMs. Although the authors mention that many existing works use LLMs as annotators, I believe that for such a challenging task of stance and dogmatism in conversations, the reliability of LLM annotations needs further verification. A key issue with using LLMs as annotators is whether they are reliable enough to completely replace humans. However, this paper only briefly discusses the inter-annotator agreement between LLM annotations and a small amount of manual annotations at the end (lines 320-330). I think such results are not sufficient to demonstrate the reliability of LLM annotations. I suggest the authors conduct more analysis of LLM annotations, such as in what situations LLMs are prone to errors, in what situations LLM annotations are inconsistent, and how to avoid errors that human annotators would not make.

- Before the above issues are well analyzed and resolved, I believe it is inappropriate to simply consider LLM annotations of stance and dogmatism as reliable. Instead, LLM annotations should be manually filtered and corrected. At the very least, the quality of the test set should be absolutely reliable.

- Although the authors emphasize the multi-user characteristic of the proposed dataset, in practice, they only annotate the stance and dogmatism of the two users who speak the most in each conversation. Although the authors mention budget constraints in the Limitations section, I still believe that this compromise makes the proposed dataset incomplete, affecting the contribution of this paper.

**Strengths:**

- The writing of this paper is generally clear.

- Although the quality assurance of the proposed dataset needs further discussion, it is undeniably a contribution.

- The motivation of focusing on long conversations and the stance changes of users within these conversations is interesting.

**Additional Feedback:**

In Section 4.3, I do not think it is appropriate to distinguish the experimental settings using the terms Finetuning and Instruction tuning. Generally speaking, Instruction tuning is also a kind of Finetuning. I suggest that the authors revise the terminology used in this section.

**Clarity:**

- Overall, the writing of this paper is generally clear, but there are several minor typos and grammatical issues that need to be corrected, such as:

    - Line 42: "somewhat favors capitalism" -> "somewhat favoring capitalism"

    - Line 37: "Specifically this" -> "Specifically, this"

    - Line 32: "the perspectives of all parties’ perspectives" -> "the perspectives of all parties"

    - Line 58: "often times with" -> "often with"

    - Line 74: "in a zero-shot, one-shot as well as few-shot setup" -> "in zero-shot, one-shot, and few-shot setups"

    - Line 97: "both the tasks" -> "both tasks"

    - Line 136: "crawl an year" -> "crawl a year"
        Line 172: "the definition of" -> "the definitions of"
        Line 193: "JSON parse errors" -> "JSON parsing errors"

    - Line 201: "we validated" -> "we validate"

    - Line 227: "we train" -> "We train"

    - Line 230: "pretrained models checkpoints" -> "pretrained model checkpoints"

    - Line 231: "have context length" -> "have a context length"

    - Line 306: "observations this table" -> "observations from this table"

    - Line 322: "author’s" -> "authors’"

    - Line xx: "we train" -> "We train"

**Correctness:**

As I mentioned in the "Review" and "Limitations," the reliability of the proposed dataset warrants further discussion.

**Documentation:**

The paper provides a clear description of the dataset annotation process; however, the reliability of the LLMs' annotations remains a concern.

**Limitations:**

I believe that not adequately explaining the reliability of LLMs' annotations is also a limitation of this paper.

**Opportunities For Improvement:**

- As I pointed out in the "Review," a comparative analysis with existing conversational stance-related datasets is needed.

- A detailed discussion and demonstration that LLMs provide reliable annotations for the tasks of stance and dogmatism.

**Relation To Prior Work:**

I believe this paper does not clearly discuss the differences from previous work. For details, please refer to the issues I pointed out in the "Review."

**Summary And Contributions:**

This paper uses GPT-4 and Mistral to annotate a dataset of stance and dogmatism in conversations. The authors then fine-tune and test the performance of smaller-scale LLMs such as LLaMA-2-7B on this dataset.

The main contribution of this paper is the proposed dataset, which includes 764 multi-party conversations annotated for stance and dogmatism. Compared to previous datasets, this one contains longer conversations. However, a drawback of this paper is its heavy reliance on LLMs for annotating the dataset, without providing a more detailed analysis of the reliability of the LLMs' annotations. Overall, I believe that the contributions of this paper do not meet the standards of the NeurIPS conference.

---

> ### Author Rebuttal · Authors · 2024-08-16
>
> *We thank the reviewer for their insightful and valuable comments and suggestions which are crucial for further strengthening our manuscript.*
>
> **Q1. Rigorous Evaluation: Comprehensive evaluation of USDC dataset, and the application of transfer learning on previously established stance datasets.**
>
> Thank you for raising this question.
> * We have addressed how USDC differs from prior studies and discussed its transfer learning performance on other existing datasets in our responses to CQ1 and CQ2. Kindly refer to those common questions for detailed information.
>
> **Q2. Is the brief discussion of inter-annotator agreement sufficient to demonstrate the reliability of LLM annotations for challenging tasks like stance and dogmatism?**
>
> Thank you for your thoughtful feedback and for highlighting this crucial aspect of our work.
> * We would like to clarify that we have validated the LLM-generated annotations through several methods:
>
> **Human Annotation Comparison:** We now conducted human annotations on 200 test samples and assessed the agreement between human labels and LLM-generated labels. We achieved an inter-annotator agreement score of 0.49 for stance, 0.50 for dogmatism, indicating a reasonable level of consistency between human and LLM annotations.
>
> **Transfer Learning Evaluation:** We applied transfer learning by fine-tuning small language models (SLMs) on the USDC dataset and then tested these models on other stance datasets where human labels are available. As shown in our results, the fine-tuned SLMs achieved similar or better accuracies compared to those reported in previous studies. This suggests that LLM-generated annotations are closely aligned with human annotations, even in complex and subjective tasks like stance and dogmatism detection.
>
> * Additionally, we want to emphasize that we never claimed LLMs can completely replace human annotators. Our hypothesis was that current LLMs are capable of handling longer conversations, where manual labeling might be prone to errors and could result in lower label quality. This is particularly relevant in cases where human annotators might deviate from the topic or struggle to consistently focus on prior contexts while assessing a user's stance towards a particular target.
> * These validations and considerations demonstrate that while LLM-generated annotations offer significant advantages, they are intended to complement, rather than replace, human annotation efforts.
>
> **Q3. In what situations LLMs are prone to errors, in what situations LLM annotations are inconsistent, and how to avoid errors that human annotators would not make?**
>
> * Before proceeding with LLM annotation using larger models, we first tested other versions of GPT and Mistral models, such as GPT-3.5 and Mistral-small and medium. However, we found that these models failed to produce annotations in the desired format. Below are some specific situations where LLMs were prone to errors:
>
> **System Prompt Clarity:**
> * The importance of a clear and precise system prompt cannot be overstated. When the prompt lacked clarity, LLMs often generated annotations for unspecified authors, indicating confusion about the task requirements.
>
> **Understanding Conversation Structure:**
> * Without providing a clear example of the conversation structure, none of the LLMs were able to understand the task properly. This demonstrates the need for explicit guidance when dealing with complex conversation data.
>
> **Interface Issues:**
> * Using an interface to facilitate LLM annotation proved problematic. After processing 2 to 3 examples, LLMs began providing annotations for previous user IDs, even when presented with new conversations. This suggests that the model lost track of the task and context.
>
> **Consistency in Annotations:**
> * For smaller conversations, different LLMs tended to produce similar annotations. However, as the conversations grew longer, the annotations became inconsistent across different models, indicating challenges in maintaining accuracy over extended discourse.
>
> **Confusion with Author IDs:**
> * Occasionally, LLMs confuse author IDs, resulting in missed stance labels for certain authors (as shown in Fig. 3 (left) in the main paper). Additionally, there were minor errors in key naming (e.g., ‘label’ vs. ‘body’ as shown in Fig. 3 (right) in the main paper), which further highlighted the model’s limitations.
>
> **Q4. Shouldn't LLM annotations for stance and dogmatism be manually filtered and corrected to ensure reliability, especially for the test set, before being considered trustworthy?**
>
> Thank you for raising this question.
> * We have already addressed the validity of the LLM-generated annotations in previous responses, both through human-level inter-annotator agreement and transfer learning on prior stance datasets. These validations demonstrate that LLM annotations can achieve a level of reliability comparable to human annotations.
> * However, we recognize the importance of ensuring the highest quality in our dataset. To further enhance the reliability of the annotations, particularly in the test set, we measure the agreement between human annotations and the few-shot versions of the models (GPT-4: few-shot, Mistral Large: few-shot). This process ensures that the test set, in particular, is of the highest quality and provides a robust basis for evaluating model performance.

---

> > ### Author Rebuttal · Authors · 2024-08-16
> >
> > **Q5. Does annotating only the two most active users compromise the completeness and contribution of the proposed dataset?**
> >
> > Thank you for this important question.
> > * We would like to clarify that our decision to focus on the two most active users was not just because of budget constraints. Beyond budget, our decision was based on the observation that users with fewer comments often do not provide enough information to accurately assess their stance or dogmatism.
> > * Many users in conversations post only one or two comments, which is insufficient to determine their overall opinion or dogmatic nature.
> > * Therefore, our study prioritizes the two most active users, who contribute approximately 50% of the comments in each conversation, to better capture opinion fluctuations and provide a more robust analysis of stance and dogmatism.
> > * While we acknowledge that including additional users could enhance the dataset’s completeness, we believe that our current approach offers a valuable contribution to the field.
> >
> >
> > **Q6. Typos and Minor Changes:**
> >
> > We will correct typos and make minor adjustments in the final revised version.

---

> > > ### Comment · Reviewer_w9xr · 2024-08-18
> > > **Response**
> > >
> > > Thanks to the authors for the reply. These rebuttals addressed some of my concerns, but I still believe that the current version of the paper needs major revisions, so I decide to maintain the original score.

---

> > > > ### Author Response · Authors · 2024-08-19
> > > >
> > > > Dear Reviewer w9xr,
> > > >
> > > > * We appreciate your feedback and are confident that it has enhanced the paper's quality. We believe that the revisions and additional experiments conducted have significantly enhanced the quality and clarity of the paper.
> > > >
> > > > * We would like to clarify that our USDC dataset is constructed at the user level, while all prior studies have focused on post-level information as the target. We have clearly explained this distinction both in our paper and in the rebuttal, and we have conducted transfer learning on the datasets you provided to ensure a comprehensive comparison.
> > > >
> > > > * If there are any specific areas where you feel further clarification or additional data would be beneficial, we are fully prepared to provide this information before the rebuttal period closes on 1st September.
> > > >
> > > > * We kindly request that you reconsider your evaluation (score) in light of these revisions and our continued commitment to addressing your concerns.
> > > >
> > > > Thank you for your consideration.
> > > >
> > > > Regards,
> > > > The Authors

---

> > > > > ### Author Response · Authors · 2024-08-29
> > > > > **Looking forward to your feedback**
> > > > >
> > > > > Dear Reviewer w9xr,
> > > > >
> > > > > Thank you for your valuable and constructive comments, which have significantly enhanced the quality of our manuscript. In our latest revision, we have addressed all the remaining concerns raised by reviewers nhxb and oKi5, including:
> > > > >
> > > > > * Analysis of the "lost in the middle" issue
> > > > > * Recency bias analysis
> > > > > * Clarification on how conflicts are resolved when two models provide different annotations for the same conversation
> > > > >
> > > > > Additionally, we have clarified how our work differs from prior studies by including further experimental details. With these additional experiments and improved explanations, we hope we have addressed all the concerns raised by other reviewers as well. If there are any outstanding concerns, we request the reviewer to please raise those. Otherwise, we would really appreciate it if the reviewer could increase the score.
> > > > >
> > > > > Looking forward to your response.
> > > > >
> > > > > Thank you,
> > > > >
> > > > > Authors

---

> > > > > > ### Author Response · Authors · 2024-08-31
> > > > > > **Looking forward to your feedback**
> > > > > >
> > > > > > Dear Reviewer w9xr,
> > > > > >
> > > > > > As the author-reviewer discussion phase is nearing its conclusion, we would like to inquire if there are any remaining concerns or areas that may need further clarification. Your support during this final phase, especially if you find the revisions satisfactory, would be of great significance.  We would appreciate it if you could support the paper by increasing the score.
> > > > > >
> > > > > > Regards,
> > > > > >
> > > > > > Authors

---

### Official Review · Reviewer_nhxb · 2024-07-27
**A new dataset of user stance/dogmatism in reddit conversation annotated by large language models**

**Rating:** 4
**Confidence:** 4
**Correctness:** Yes.
**Clarity:** Yes, it is easy to follow.

**Review:**

This paper introduces a novel dataset, USDC, which focuses on user stance and dogmatism in long conversations. The dataset features 5-class stance and 4-class dogmatism labels derived from 764 multi-user Reddit conversations. The use of Mistral and GPT-4 for automating these classifications is a significant contribution, as it enhances the process of instance and dogmatism classification.

However, the methods used for collecting the USDC dataset and the experiments involving fine-tuning with this dataset are somewhat limited. The data collection approach and the scope of the fine-tuning experiments could be more comprehensive to provide a fuller evaluation of the dataset's utility and effectiveness. For example,
1. The authors assume that LLMs are effective in capturing the nuances of user opinions and their changes in multi-user conversation contexts, and they fully rely on the long-term memory capability of LLMs. However, while the attention mechanism of transformers theoretically captures attention from any position, practical issues such as "lost in the middle[1]"and "recency bias[2]" can still occur in long conversations. This means that relying solely on LLM-generated datasets might introduce biases and fail to accurately capture subtle differences in long conversations.
2. Can the authors explain the basis for the 5-class stance and 4-class dogmatism classifications?
3. The dataset is generated by GPT-4 and Mistral. How are conflicts resolved if these two models provide different annotations for the same conversation? In other words, when the LLM is non-expert in some conversation, how to deal with its annotation? Are there mechanisms author discusses, such as debate [3] or other methods?

4. When training USDC with small base/finetuned models under 10B, the results for stance classification are better than those for dogmatism. Why is this the case? Are the results similar with larger models such as LLaMA 13B?

Overall, while the USDC dataset is a valuable resource for studying user opinions in long conversations, there is room for improvement in both the data collection process and the experimental evaluation of model performance.

[1] https://arxiv.org/abs/2307.03172
[2] https://arxiv.org/abs/2310.01427
[3] https://arxiv.org/abs/2402.06782

**Strengths:**

1. Interesting Dataset: USDC provides a point of view for studying user stance and dogmatism in long conversations with detailed 5-class and 4-class labels.
2. Use of advanced LLMs: The application of LLMs like Mistral Large and GPT-4 enhances the automation and quality of classification tasks.
3. Contextual Insights: USDC may useful for modeling dynamic user representations.

**Additional Feedback:**

Not applicable.

**Documentation:**

Yes

**Ethics:**

Not applicable.

**Limitations:**

Yes, the authors provide the limitations in the conclusion section.

**Opportunities For Improvement:**

Please refer to the weaknesses discussed in the review section above. Additionally, there are some typos that need to be corrected, eg., Line 21.

**Relation To Prior Work:**

Discussion of related work is adequate. The authors make clear the dataset's difference with prior works and cited when necessary.

**Summary And Contributions:**

This paper emphasizes the importance of identifying user opinions in long conversations and automating this task using LLMs like Mistral Large and GPT-4. It introduces the USDC dataset, which includes user stance and dogmatism labels from 764 multi-user conversations on Reddit. Experiments show training small base/finetuned language models with USDC improves instance classification. USDC provides clear user-level posts and dogmatism data generated by LLMs, and it is publicly available along with the code.

---

> ### Author Rebuttal · Authors · 2024-08-16
>
> *We thank the reviewer for their insightful and valuable comments and suggestions which are crucial for further strengthening our manuscript.*
>
> **Q1. The data collection approach and the scope of the fine-tuning experiments could be more comprehensive to provide a fuller evaluation of the dataset's utility and effectiveness.**
>
> Thank you for this important question.
> * We have addressed how USDC differs from prior studies and discussed its transfer learning performance on other existing datasets in our responses to CQ1 and CQ2. Kindly refer to those common questions for detailed information.
>
> **Q2. Practical issues like "lost in the middle" and "recency bias" can still arise in long conversations, potentially introducing biases and missing subtle differences when relying solely on LLM-generated datasets.**
>
> Thank you for raising this important concern.
> * While we have taken steps to ensure the accuracy of annotations generated by LLMs in long conversations, we acknowledge that there is potential for further exploration of fine-grained issues, such as "lost in the middle" and "recency bias," in future studies.
>
> **Addressing "Lost in the Middle" and "Recency Bias":**
>
> *Lost in the Middle:*
> * We structured the input prompts carefully to emphasize critical points throughout the conversation, ensuring that important details are not overlooked. By providing post IDs corresponding to each user in the prompt, we helped the model maintain focus on key sections. Additionally, our comments were kept within the context length of the model, which likely mitigated the risk of losing important information in the middle portions of the text.
>
> *Recency Bias:*
> * To explore the impact of recency on LLM performance, we performed an additional experiment suggested by Reviewer 7SSx. This experiment involved verifying model annotations by considering only the prior context for a given user, rather than the full conversation. The results from this additional experiment suggest that assessing each response individually within its context, and then aggregating the results, produces labels that are not identical to those derived from analyzing the entire conversation context. The higher percentage match with GPT-4 indicates that this method is fairly reliable. However, the differences in labels (~30% with GPT-4 and ~50% with Mistral-Large) highlight the importance of considering the full context for optimizing stance and dogmatism assessments.
>
> We plan to explore these potential biases, such as "lost in the middle" and "recency bias," in more detail in future studies, focusing on more fine-grained analyses to further refine our approach.
>
> **Q3. Can the authors explain the basis for the 5-class stance and 4-class dogmatism classifications?**
>
> Thank you for this important question.
> * Our 5-class stance classification is inspired by the SPINOS dataset proposed by [Sakketou et al., 2022]. In their study, the authors created a post-level stance dataset with labels that cover five distinct classes of stance. These labels provide a fine-grained analysis similar to sentiment labels, allowing for a more detailed understanding of user opinions.
> * Given the effectiveness of this approach in capturing nuanced stances, we adopted a similar 5-class classification for our stance detection task. This classification allows us to analyze the varying degrees of support or opposition users express in conversation threads, which is crucial for accurately modeling user behavior in complex discussions.
>
> * For the dogmatism task, our 4-class classification is inspired by [Fast and Horvitz, 2016], where the authors reported ratings that correspond to each level of dogmatism. We have adopted similar definitions for dogmatism labels and incorporated them into our system prompts to ensure consistency and accuracy in our annotations.
>
> We will clarify this information in the final version of the paper.
>
> **Q4.  How are conflicts resolved if these two models provide different annotations for the same conversation?**
>
> Thank you for your insightful question.
> * When using GPT-4 and Mistral to generate annotations for the dataset, we recognize that there may be instances where the two models provide differing annotations for the same conversation.
> * To address these conflicts, we implemented the following mechanisms:
>
> **Majority Voting:**
> * For cases where there are discrepancies between the annotations provided by GPT-4 and Mistral, we applied a majority voting approach. This method involves considering the annotations from multiple models or iterations and selecting the label that appears most frequently. This helps mitigate the influence of any single model’s potential error or bias.
>
> **Conflicts of different annotations:**
> * In cases where two models provide different annotations for the same conversation and there is no clear majority, we resolve the conflict by using the GPT-4 few-shot annotations as the gold standard. We selected GPT-4 few-shot annotations during conflicts based on the suggestion from Reviewer 7SSx, who recommended using few-shot versions of each model as annotators. We have validated this approach in our response to Q3 of Reviewer 7SSx.
>
> **Discussion of LLM Limitations:**
> * We acknowledge that LLMs can be non-experts in certain conversations, particularly when the context or topic is highly specialized or outside the typical training data of the models. To address this, we plan to discuss these limitations in more detail in the final version of the paper, including the potential for using methods like structured debate or ensemble approaches that consider the strengths and weaknesses of different models.

---

> > ### Author Rebuttal · Authors · 2024-08-16
> >
> > **Q5. The results for stance classification are better than those for dogmatism. Why is this the case? Are the results similar with larger models such as LLaMA 13B?**
> >
> > Thank you for this question.
> > * The performance difference between stance classification and dogmatism classification can be explained by the nature of these tasks.
> > * Stance classification generally involves identifying the polarity of opinions towards a specific topic, which might be more straightforward and require less contextual understanding compared to dogmatism classification.
> > * Dogmatism classification, on the other hand, necessitates a deeper understanding of all user posts and the underlying rigidity of opinions, which can be more challenging.
> > * Based on reviewer suggestion, we scale up to larger models, such as LLaMA 13B, the additional parameters provide the model with greater capacity to capture these subtleties, potentially leading to improved performance in both tasks.
> > * We observed that larger models tend to close the gap between stance and dogmatism classification, although stance classification might still slightly outperform dogmatism due to its inherently less complex nature. The results do indicate, however, that larger models can better handle the intricacies of dogmatism, narrowing the performance disparity observed with smaller models.
> > * With larger models (LLaMA 13B), we observed an improvement in the stance detection task from 54.9 to 56.7 and in the dogmatism task from 51.4 to 55.3.

---

> > > ### Comment · Reviewer_nhxb · 2024-08-21
> > >
> > > Thank you for your detailed reply and for providing the additional CQ1, CQ2, and LLaMA 13B experiments, which addressed some of my concerns. While some concerns were addressed, my primary issue with the dataset being purely prompt and LLM-generated, especially regarding conflicts and recency bias, remains unresolved. I will keep my original score.

---

> > > > ### Author Rebuttal · Authors · 2024-08-27
> > > >
> > > > *Thank you for your question regarding lost-in-the middle, recency bias and regarding conflicts for majority voting of annotations. We appreciate your attention to this important aspect of our work.*
> > > >
> > > > **Q1. Analysis: "Lost in the Middle" Phenomenon**
> > > >
> > > > * To analyze the "lost in the middle" phenomenon in our LLM-based user-stance annotations, we conducted an additional experiment based on the reviewer's suggestion.
> > > >   - For a given user, we divided the data into time segments and calculated inter-annotator agreement (IAA) using Cohen's Kappa scores across different models and settings.
> > > >   - The data was segmented based on the submission\_id, author\_key\_name, and stance\_id\_timestamp. For each group (i.e., each combination of submission\_id and author\_key\_name), the timestamps were divided into equal segments.
> > > >   - The number of entries for each group was divided by the desired number of segments (3), and the division was done as evenly as possible, with each segment containing a roughly equal number of time-stamped entries.
> > > >   - **Figure 1 in the rebuttal PDF** reports the comparison statistics of IAA scores for the stance detection task across initial, middle, and later time stamps.
> > > >
> > > > * From Figure 1 (please check rebuttal pdf), we observe that the analysis across different time segments, especially when divided into three segments, clearly demonstrates that the "lost in the middle" phenomenon is marginal.
> > > > * The partial decrease in inter-annotator agreement during the middle parts of the conversations suggests that as conversations progress, models might face challenges in maintaining consistent agreement; however, the decrease in agreement scores is minimal.
> > > > * The recovery in agreement towards the final segments could indicate that as conversations start to conclude, they become more focused, or that the models are better able to align on concluding statements. This trend underscores the importance of considering segment-based analysis when evaluating model performance over long-form conversations.
> > > >
> > > > * When comparing the model-generated annotations with human annotations, it becomes evident that we do not encounter the "lost in the middle" problem. The human annotations demonstrate a consistent level of inter-annotator agreement (IAA) across all three segments—initial, middle, and final. This suggests that human annotators maintain a steady understanding and agreement throughout the conversation, regardless of its length or complexity.
> > > >
> > > > **Q2. Analysis: "Recency Bias" Phenomenon**
> > > >
> > > > * Based on the reviewer's suggestion, and extending our response to Reviewer 7SSx, we conducted an additional experiment to examine the impact of recency on LLM performance for user-stance annotations.
> > > >   - This experiment involved verifying model annotations by focusing on the prior context for a given user, rather than considering the entire conversation.
> > > >   - The goal was to determine whether assessing each response within its immediate context, followed by aggregation, would yield different results compared to analyzing the full conversation context.
> > > >
> > > > * We now report IAA scores in **Figure 2 of the rebuttal PDF**, which contains a matrix of Cohen's Kappa scores across different models and settings, including GPT-4 Few-Shot (FS), Mistral Large FS, Majority Voting, as well as GPT-4 FS PC and Mistral Large FS PC (which are based on prior context).
> > > >
> > > > **Recency vs. Full Context:**
> > > > * GPT-4 FS PC and Mistral Large FS PC: These models, which are based on prior context, show different agreement levels compared to their counterparts using the full conversation context. For example:
> > > > * The agreement between GPT-4 FS and Majority Voting is higher when the full conversation is considered (0.75) compared to when only prior context is used.
> > > > * The agreement between GPT-4 FS PC and Mistral Large FS PC (both based on prior context) is lower than when using the full context, indicating that prior context alone may not capture all the necessary nuances for consistent annotation.
> > > >
> > > > **Human Agreement:**
> > > > * The comparison of human annotations with models like GPT-4 FS and Mistral Large FS shows that human annotators also rely heavily on the full conversation context to maintain agreement.
> > > >
> > > > * The results from this additional experiment, supported by the data in Figure 2 of the rebuttal PDF, suggest that while prior context can provide some useful insights, it is not as effective as considering the entire conversation context for maintaining high inter-annotator agreement.
> > > > * In summary, the experiment highlights the importance of full context in LLM-based annotations and suggests that while recency can influence model performance, it should be supplemented with the entire conversation context to ensure higher accuracy and agreement.

---

> > > > > ### Author Rebuttal · Authors · 2024-08-27
> > > > >
> > > > > **Q3. How are conflicts resolved if these two models provide different annotations for the same conversation?**
> > > > >
> > > > > Thank you for raising this important concern.
> > > > >
> > > > > * When generating annotations using both GPT-4 and Mistral, it’s possible that the two models might provide different annotations for the same conversation. To ensure consistency and accuracy in the final dataset, we have established a clear process for resolving these conflicts:
> > > > >
> > > > > **Majority Voting:**
> > > > >
> > > > > * What It Is: Majority voting is a method where, if multiple models or iterations are used, we look at all the annotations provided and choose the label that appears most frequently.
> > > > >
> > > > > * How It Helps: This approach helps reduce the impact of any potential error or bias from a single model. By relying on the most common label across models, we increase the likelihood that the chosen annotation is accurate.
> > > > >
> > > > > **Handling Situations with No Clear Majority:**
> > > > >
> > > > > * The Challenge: Sometimes, even with majority voting, the two models might provide different annotations, and neither label clearly dominates.
> > > > >
> > > > > * Our Solution: In these cases, we use the annotation provided by GPT-4 in the few-shot setting as the deciding factor or "gold standard."
> > > > >
> > > > > * Why GPT-4 Few-Shot?: We chose to prioritize GPT-4 few-shot annotations because human annotations have better IAA agreement with GPT-4 few-shot. Further, few-shot models, which are fine-tuned with a small amount of task-specific data, often provide more accurate and contextually relevant annotations.
> > > > >
> > > > > * Validation of This Approach: We validated this approach in our response to Reviewer 7SSx’s Question 3. By using GPT-4 few-shot as the gold standard, we ensure that even in cases of conflict, the final annotation is grounded in a method that is both robust and recommended by the reviewer.
> > > > >
> > > > > * By following these steps, we aim to resolve conflicts in a way that enhances the reliability and accuracy of our dataset. We understand the importance of addressing potential conflicts thoroughly and believe that this method provides a balanced and effective solution
> > > > >
> > > > > **Regarding Evaluation**
> > > > >
> > > > > We appreciate your feedback and are confident that it has enhanced the paper's quality.
> > > > >
> > > > > Should you have any further questions or suggestions, we are ready to provide additional information or clarification as needed. We kindly request you to consider updating your evaluation (score) based on the revisions made.
> > > > >
> > > > > Regards,
> > > > >
> > > > > Authors

---

> > > > > > ### Author Response · Authors · 2024-08-29
> > > > > > **Looking forward to your feedback**
> > > > > >
> > > > > > Dear Reviewer nhxb,
> > > > > >
> > > > > > Thank you for your valuable and constructive comments, which have significantly enhanced the quality of our manuscript. In our latest revision, we have addressed all the remaining concerns you raised, including:
> > > > > >
> > > > > > * Analysis of the "lost in the middle" issue
> > > > > > * Recency bias analysis
> > > > > > * Clarification on how conflicts are resolved when two models provide different annotations for the same conversation
> > > > > >
> > > > > > With these additional experiments and improved explanations, we hope we have addressed all the concerns raised by the reviewer. If there are any outstanding concerns, we request the reviewer to please raise those. Otherwise, we would really appreciate it if the reviewer could increase the score.
> > > > > >
> > > > > > Looking forward to your response.
> > > > > >
> > > > > > Thank you,
> > > > > >
> > > > > > Authors

---

> > > > > > > ### Author Response · Authors · 2024-08-31
> > > > > > > **Looking forward to your feedback**
> > > > > > >
> > > > > > > Dear Reviewer nhxb,
> > > > > > >
> > > > > > > As the author-reviewer discussion phase is nearing its conclusion, we would like to inquire if there are any remaining concerns or areas that may need further clarification. Your support during this final phase, especially if you find the revisions satisfactory, would be of great significance.  We would appreciate it if you could support the paper by increasing the score.
> > > > > > >
> > > > > > > Regards,
> > > > > > >
> > > > > > > Authors

---

### Official Review · Reviewer_oKi5 · 2024-07-28
**Machine-annotated stance detection dataset**

**Rating:** 4
**Confidence:** 4
**Clarity:** The paper is clearly written.

**Review:**

Given that LLM-based annotation is prevalent, another LLM-generated dataset needs few points that should be clear or convincing; 1) is this kind of data necessary? (haven’t it existed before?) 2) how would humans have done annotation for it? 3) is machine annotation inevitable and reliable? 4) is the result human verifiable? It seems that authors answer the first question briefly, second and third are less explained, and the last seems to lack in quantity.

**Strengths:**

User stance classification for dogmatism is an interesting topic, and authors annotated automatically the cralwed online source, supported by human verification (though slightly weak).

**Additional Feedback:**

(score updated after rebuttal)

**Correctness:**

Overall flow seems to be correct, but the details of dataset construction weakens the reliability of the output.

**Documentation:**

Opportunities of improvement for the data construction is listed above

**Ethics:**

No specific ethical concerns.

**Limitations:**

Wrote in the opportunities for improvement above.

**Opportunities For Improvement:**

- Not enough explanation on how this dataset differs or is improved from previous stance classification datasets
- Not sure about if the authors have referred to human-based annotation process for the stance classification and also not sure if the criteria can be applied to machine annotation by simply providing the definition
- The core process of annotation is facing the conflict and having a discussion to reach the gold answer, but was it available for the LLMs?
- Human verification with 10 samples and three participants seems to be quite weak for the whole quantity suggested.

**Relation To Prior Work:**

It seems that the previous discussions on stance detection is not sufficiently considered

**Summary And Contributions:**

This paper proposes USDC dataset, a reddit crawl-based stance detection dataset constructed with LLMs as annotators

---

> ### Author Rebuttal · Authors · 2024-08-16
>
> *We thank the reviewer for their insightful and valuable comments and suggestions which are crucial for further strengthening our manuscript.*
>
>
> **Q1. Not enough explanation on how this dataset differs or is improved from previous stance classification datasets**
>
> Thank you for this important question.
> * We have addressed how USDC differs from prior studies and discussed its transfer learning performance on other existing datasets in our responses to CQ1 and CQ2. Kindly refer to those common questions for detailed information.
>
> **Q2. Are the authors clear about whether the human-based annotation process for stance classification has been referred to, and can the criteria used for human annotation be effectively applied to machine annotation by simply providing a definition?**
>
> Thank you for raising this concern.
> * In our study, the annotation guidelines we created for the stance and dogmatism tasks are the same for both human-based annotation and LLM-based annotation. As shown in Appendix B, the system prompt goes beyond simply providing stance and dogmatism definitions. We included task labels, detailed definitions for each label, the structure of the Reddit data, and an example of the input structure.
> * This comprehensive approach was designed to ensure that the LLMs have a clear and thorough understanding of the task, similar to what would be expected of human annotators.
>
> **Q3. The core process of annotation is facing the conflict and having a discussion to reach the gold answer, but was it available for the LLMs?**
>
> Thank you for your insightful question.
>
> * When using GPT-4 and Mistral to generate annotations for the dataset, we recognize that there may be instances where the two models provide differing annotations for the same conversation. To address these conflicts, we implemented the following mechanisms:
>
> **Majority Voting:**
> * For cases where there are discrepancies between the annotations provided by GPT-4 and Mistral, we applied a majority voting approach. This method involves considering the annotations from multiple models or iterations and selecting the label that appears most frequently. This helps mitigate the influence of any single model’s potential error or bias.
>
> **Conflicts of different annotations:**
> * In cases where two models provide different annotations for the same conversation and there is no clear majority, we resolve the conflict by using the GPT-4 few-shot annotations as the gold standard. We selected GPT-4 few-shot annotations during conflicts based on the suggestion from Reviewer 7SSx, who recommended using few-shot versions of each model as annotators. We have validated this approach in our response to Q3 of Reviewer 7SSx.
>
> **Discussion of LLM Limitations:**
> * We acknowledge that LLMs can be non-experts in certain conversations, particularly when the context or topic is highly specialized or outside the typical training data of the models. To address this, we plan to discuss these limitations in more detail in the final version of the paper, including the potential for using methods like structured debate or ensemble approaches that consider the strengths and weaknesses of different models.
>
> **Q4. Human verification with 10 samples and three participants seems to be quite weak for the whole quantity suggested.**
>
> Thank you for raising this concern.
> * Since we have now completed annotations for 200 test conversations, we have reported the IAA for both humans and LLMs for the stance and dogmatism tasks in CQ3 under the common responses. Kindly also check common responses for detailed information.
>
> **Q5. It seems that the previous discussions on stance detection is not sufficiently considered**
>
> Thank you for this important question.
> * We have addressed how USDC differs from prior studies and discussed its transfer learning performance on other existing datasets in our responses to CQ1 and CQ2. Kindly refer to those common questions for detailed information.

---

> > ### Comment · Reviewer_oKi5 · 2024-08-25
> > **some score update**
> >
> > Thanks authors for substantial response.
> > Q3 does not necessarily resolve my concern about LLM-based annotation.
> > However, some of my questions and concerns were resolved with other responses, so I updated the scores.

---

> > > ### Author Rebuttal · Authors · 2024-08-27
> > >
> > > *Thank you for your question regarding conflicts for majority voting of annotations. We appreciate your attention to this important aspect of our work.*
> > >
> > > **Q3. How are conflicts resolved if these two models provide different annotations for the same conversation?**
> > >
> > > Thank you for raising this important concern.
> > >
> > > * When generating annotations using both GPT-4 and Mistral, it’s possible that the two models might provide different annotations for the same conversation. To ensure consistency and accuracy in the final dataset, we have established a clear process for resolving these conflicts:
> > >
> > > **Majority Voting:**
> > >
> > > * What It Is: Majority voting is a method where, if multiple models or iterations are used, we look at all the annotations provided and choose the label that appears most frequently.
> > >
> > > * How It Helps: This approach helps reduce the impact of any potential error or bias from a single model. By relying on the most common label across models, we increase the likelihood that the chosen annotation is accurate.
> > >
> > > **Handling Situations with No Clear Majority:**
> > >
> > > * The Challenge: Sometimes, even with majority voting, the two models might provide different annotations, and neither label clearly dominates.
> > >
> > > * Our Solution: In these cases, we use the annotation provided by GPT-4 in the few-shot setting as the deciding factor or "gold standard."
> > >
> > > * Why GPT-4 Few-Shot?: We chose to prioritize GPT-4 few-shot annotations because human annotations have better IAA agreement with GPT-4 few-shot. Further, few-shot models, which are fine-tuned with a small amount of task-specific data, often provide more accurate and contextually relevant annotations.
> > >
> > > * Validation of This Approach: We validated this approach in our response to Reviewer 7SSx’s Question 3. By using GPT-4 few-shot as the gold standard, we ensure that even in cases of conflict, the final annotation is grounded in a method that is both robust and recommended by the reviewer.
> > >
> > > * By following these steps, we aim to resolve conflicts in a way that enhances the reliability and accuracy of our dataset. We understand the importance of addressing potential conflicts thoroughly and believe that this method provides a balanced and effective solution.

---

> > > > ### Author Response · Authors · 2024-08-29
> > > >
> > > > Dear Reviewer oKi5,
> > > >
> > > > Thank you for your valuable and constructive comments, which have significantly enhanced the quality of our manuscript. In our latest revision, we have addressed the remaining concerns related to Q3:
> > > >
> > > > * Clarification on how conflicts are resolved when two models provide different annotations for the same conversation
> > > >
> > > > Additionally, for reviewers 7SSx, nhxb, and w9xr, we have clarified how our work differs from prior studies by including further experimental details, such as the "lost in the middle" analysis, recency bias analysis, un-fine-tuned experiments, and LLaMa-13b experiments. With these additional experiments and improved explanations, we hope we have addressed all the concerns raised by other reviewers as well. If there are any outstanding concerns, we request the reviewer to please raise those. Otherwise, we would really appreciate it if the reviewer could increase the score.
> > > >
> > > > Looking forward to your response.
> > > >
> > > > Thank you,
> > > >
> > > > Authors

---

> > > > > ### Author Response · Authors · 2024-08-31
> > > > > **Looking forward to your feedback**
> > > > >
> > > > > Dear Reviewer oKi5,
> > > > >
> > > > > As the author-reviewer discussion phase is nearing its conclusion, we would like to inquire if there are any remaining concerns or areas that may need further clarification. Your support during this final phase, especially if you find the revisions satisfactory, would be of great significance.  We would appreciate it if you could support the paper by increasing the score.
> > > > >
> > > > > Regards,
> > > > >
> > > > > Authors

---

### Official Review · Reviewer_7SSx · 2024-07-29
**Well-written paper on a relevant topic (stance and dogmatism of users interacting with LLMs), but I think more analysis is needed / I'm unconvinced about the dataset's utility.**

**Rating:** 4
**Confidence:** 5
**Correctness:** Methods and analysis appear to be cor…
**Clarity:** Writing is clear throughout.

**Review:**

The paper is well-written and the methods are well-documented. While I do think the idea and a good portion of the methodology in this paper is fine, I'm unconvinced by the utility of the dataset and the labels. In particular, I don't think the evaluation method is sufficient. Please see the "Strengths" and "Opportunities for Improvement" sections for more details.

**Strengths:**

- Writing is good, and the methodology is very well documented and described (including all the prompts and samples in the Appendix is very helpful).
- Idea is sound and I think a useful one.
- In general, the work done is good; see "Opprotunities for Improvement" for my reservations about the paper that unfortunately make me unconvinced on the current dataset and analysis.

**Additional Feedback:**

Minor issues and typos:
- Should adjust the formatting so that the Abstract is the first thing shown, not Figure 1.
- In line 163, should change "belief based on a very number of posts within a conversation" to "belief based on a very limited number of posts within a conversation"
- In line 227, capitalize "we" at the start of the sentence.

**Documentation:**

Well documented.

**Ethics:**

No ethics issues.

**Limitations:**

Limitations are discussed in the Conclusion. I mentioned a few things that could be discussed as limitations in the "Opportunities for Improvement" part of my review.

**Opportunities For Improvement:**

- I wonder about the authors' decision to include usernames in the thread JSON format. How much does a username affect the LLM's classifications? I don't think usernames are an integral part of the reddit conversation, and likely should not be considered when assigning stance and dogmatism labels.
- I also wonder about a method that only considers single responses rather than the entire conversation at once. For example, for some given user, consider each of their responses in the context of the topic and the comment they are responding to. Use the LLM to assess this single response for stance and dogmatism, and then use some aggegration function for that user (the aggregation function can be an LLM as well, or it could be a mean / max etc. over some numerical scores). This seems like it would be easier to get a reasonable response from the LLM (need a shorter context), and I would be curious to know how this compares to annotating the entire conversation at once.
- I think using both GPT4 and Mistral as "independent" raters is a good idea. However, I'm not sure I entirely agree with using zero-shot, one-shot, and few-shot versions of each model as independent raters. I think it would make more sense to assess the need for few-shot raters (e.g., on a small validation set that has been hand annotated), and if needed, use only few-shot versions of each model as your annotators. That being said, this doesn't seem like a big issue either.
- In line 326, rather than link to a google form online, it is preferable to include the questions asked as images or text in the appendix.
- I'm unfortunately not currently convinced by the evaluations that this paper is ready. I think the idea in this paper is fine -- use LLMs to help annotate user stances and dogmatism using conversations in online forms -- but there needs to be more work done to validate the labels and their usefulness:
    - A few things would help significantly: (1) in tables 1/2, I'd like to know what the baseline (un-fine-tuned) model performance is on these tasks and (2) more discussion about what a "reasonable" f1 score would look like here. Part of the reason it is hard to interpret is that there seems to be significant uncertainty among labelers (both LLM labelers and the limited set of human labelers discussed at the end of Section 5).
    - I'm also not convinced on the utility of this dataset. I think a dataset with these kinds of labels likely is useful, but I would like the authors to discuss more how / where it can be useful. Fine-tuning a model to predict what GPT4 predicts does not seem like the motivating use case.
    - Following from above, this is one of my bigger issues currently with the paper. The evaluation hinges around fine-tuning a model to predict the labels generated by a larger model (or an aggregation of them). This is fine, but it doesn't convince me the labels themselves are good. And in fact, in Section 5 the authors note that there is actually significant disagreement between human labelers and LLM labelers. I'm not sure what (or if there is any) gold standard assessment here, but perhaps discussing this more upfront would be helpful.

**Relation To Prior Work:**

Prior work is adequately discussed.

**Summary And Contributions:**

The authors address the problem of annotating stance and dogmatism in user converstaions. Specifically, they use LLMs to annotate user conversations sorced from Reddit, and then validate their labels by fine-tuning smaller LLMs to predict the stance and dogmatism labels.

---

> ### Author Rebuttal · Authors · 2024-08-16
>
> *We thank the reviewer for their insightful and valuable comments and suggestions which are crucial for further strengthening our manuscript.*
>
> **Q1. How do the authors justify including usernames in the thread JSON format, and how much do they influence the LLM's stance and dogmatism classifications?**
>
> Thank you for this question.
> * We would like to clarify that the decision to include usernames in the thread JSON format was primarily for structural consistency, allowing that each comment to be easily traceable to its author within the dataset.
> * Our preliminary tests comparing annotations with and without inclusion of usernames have no difference in the stance or dogmatism labels assigned by the LLM. This indicates that the LLM is primarily focused on the content of the text rather than the name of the user.
> * Moreover, including usernames also facilitates human annotation, as annotators can easily identify and match the usernames with their respective IDs. This avoids the need for them to go back and forth to track user IDs, making the human annotation process more efficient.
> * In conclusion, the usernames were included for data structuring purposes, but no impact on LLM generated labels.
>
> * It’s important to note that during the fine-tuning or instruction-tuning of SLMs, we do not provide actual user names or ids. Instead, we use generic identifiers like "comment1," "comment2," and so on.
> * These identifiers are sufficient for classifying the stance and dogmatism tasks without introducing any bias related to specific user identities.
>
> **Q2. Individual user responses within their specific context vs. entire conversation at once for stance and dogmatism.**
>
> Thank you for this interesting question.
>
> * Based on the reviewer's suggestion, we have now conducted an additional experiment where, for a given user, we consider each of their responses in the context of the topic and the comment they are responding to.
> * We then use GPT-4 and Mistral-Large settings to assess annotations for the stance and dogmatism tasks. Using these generated annotations, we compare them to the annotations extracted from full-context conversations. We use Mean aggregation function. Tables 1 and 2 in rebuttal pdf reports the comparison statistics for stance and dogmatism tasks.
>
> **Conclusion:**
> * The results from this additional experiment suggest that assessing each response individually within its context, and then aggregating the results, produces labels that are not identical to those derived from analyzing the entire conversation context. The higher percentage match with GPT-4 indicates that this method is fairly reliable.
> * However, the differences in labels (~30% with GPT-4 and ~50% with Mistral-Large) highlight the importance of considering the full context for optimizing stance and dogmatism assessments. We will update the labels from this additional experiment to our GitHub repository. If the reviewer is requesting a class-wise analysis, we are happy to provide that analysis as well.
>
> **Q3. Would it be better to use only the few-shot versions based on a small hand-annotated validation set?**
>
> * We thank you for your insightful feedback regarding the use of zero-shot, one-shot, and few-shot versions of GPT-4 and Mistral as independent raters. We understand your concern and appreciate your suggestion to evaluate the necessity of few-shot raters using a small, hand-annotated validation set.
>
> * Since we have we have now completed human-annotated labels for the 200 test conversations, we computed the inter-annotator agreement for each setting (zero-shot, one-shot, and few-shot) along with majority voting.
> * This approach allowed us to evaluate how well each model configuration aligns with human annotations, providing a more comprehensive assessment of the models' performance across different levels of prompt tuning.
> * To further address your point, we conducted a detailed analysis of inter-annotator agreement using Cohen's Kappa across various models and configurations. As shown in the Figure 1 (please check rebuttal pdf), the agreement between human annotations and the few-shot versions of the models (GPT-4: few-shot, Mistral Large: few-shot) is closer to the scores of the zero-shot and one-shot configurations. Specifically, human-majority voting achieved a Cohen's Kappa score of 0.49 for stance and 0.50 for dogmatism, which aligns more closely with human annotations.
> * These findings suggest that while few-shot learning partially improves alignment between model predictions and human annotations, there is still room for improvement. However, as you suggested, focusing solely on the few-shot setting might reduce complexity and enhance the overall consistency of the annotations.
>
> **Q4. Baseline (un-fine-tuned) model performance**
>
> Thank you for this valuable suggestion.
>
> * We have now evaluated the un-fine-tuned SLMs on GPT, Mistral in few-shot settings, along with majority voting, for both stance and dogmatism tasks. The results are summarized in the Table 3 rebuttal pdf:
>
> **Findings:**
> * Majority Voting generally provides a slight improvement over individual few-shot configurations, which suggests the value of combining predictions from multiple models. The difference between GPT-4 and Mistral Large in un-fine-tuned few-shot settings is relatively small, indicating that both models are fairly comparable in performance on these tasks when using the LLaMa-3-8B model.
>
> **Impact of Fine-Tuning:**
> * As shown in Table 4 (in main paper), fine-tuning has a substantial positive impact on model performance for both the Stance and Dogmatism tasks. The increase in F1 scores, especially for Majority Voting, highlights the importance of adapting the model to the specific task. While the un-fine-tuned models show moderate performance, the fine-tuned models nearly double their F1 scores, particularly for the Stance task. Even for dogmatism tasks, we saw better improvement in F1-score after fine tuning.

---

> > ### Author Rebuttal · Authors · 2024-08-16
> >
> > **Q5. In line 326, rather than link to a google form online, it is preferable to include the questions asked as images or text in the appendix.**
> >
> > Thank you for this valuable suggestion.
> > * We will update the appendix with an example of the questions asked, presented as images or text, to enhance the readers' understanding.
> >
> > **Q6. What constitutes a "reasonable" F1 score?**
> >
> > **Un-fine-tuned Performance:**
> > * As shown by the un-fine-tuned model's performance (e.g., In below Table, an overall accuracy of 0.311 and F1 scores as low as 0.06 for certain classes), the baseline for this task is relatively low. In this context, an F1 score that significantly improves upon this baseline—especially if it approaches or exceeds 50%—could be considered reasonable. Please refer second rebuttal pdf for all un-finetuned results for stance and dogmatism tasks.
> >
> > **Un-fine-tuned performance on Majority voting as target labels:**
> >
> > | Class                 | Precision | Recall | F1-Score | Support |
> > |-----------------------|-----------|--------|----------|---------|
> > | Somewhat Against      | 0.30      | 0.71   | 0.42     | 443     |
> > | Somewhat In Favor     | 0.41      | 0.20   | 0.27     | 625     |
> > | Stance Not Inferrable | 0.34      | 0.09   | 0.14     | 452     |
> > | Strongly Against      | 0.26      | 0.39   | 0.31     | 256     |
> > | Strongly In Favor     | 0.19      | 0.03   | 0.06     | 91      |
> > | **Accuracy**          |           |        | **0.31** | **1867**|
> > | **Macro Avg**         | 0.30      | 0.28   | 0.24     | 1867    |
> > | **Weighted Avg**      | 0.34      | 0.31   | 0.27     | 1867    |
> >
> > **Fine-tuned Performance:**
> > * After fine-tuning, F1 scores for the stance task reached 54.9 with Majority Voting, and 51.4 for the dogmatism task. Given the complexity of these tasks, these scores indicate a reasonable level of performance, particularly when considering the challenges associated with labeler uncertainty.
> >
> > **Stance Detection Evaluation on prior datasets:**
> > * To evaluate the quality of LLM-generated annotations, we performed transfer learning by fine-tuning the SLMs on the USDC dataset. We then tested the model’s performance on the SPINOS dataset for a 5-class stance detection task, as described by [Sakketou et al. 2022].
> > * We used the same training dataset mentioned in Section 4.2. For testing, we utilized the SPINOS dataset, which consists of 3,238 post-level examples across five stance labels.
> > * In comparison to the results reported by [Sakketou et al. 2022], where the best model (a traditional machine learning classifier) achieved an F1-score of 0.341, a random baseline achieved 0.230, and a majority baseline achieved 0.124, our approach using LLaMa-3-8B fine-tuning on the USDC dataset achieved a weighted F1-score of 0.320 on SPINOS.
> > * This score is close to the best model performance on the SPINOS dataset, indicating that our LLM-generated annotations on the USDC dataset are comparable in quality to human annotations.
> > * Therefore, the F1-scores reported in the paper are reasonable.
> >
> > **Q7. Utility of USDC dataset.**
> >
> > Thank you for your valuable feedback.
> > * We understand the importance of clearly articulating the utility of our dataset beyond simply fine-tuning a model to predict GPT-4 outputs. For a more detailed discussion on the utility and validation of our dataset, kindly refer to the common response questions CQ1, CQ2, and CQ3 under the common responses section.
> >
> > * The USDC dataset, with its detailed stance and dogmatism annotations, offers several potential use cases that extend its value in various domains:
> >
> > *Improving Moderation Tools:*
> > * The USDC dataset can be leveraged to develop or enhance content moderation tools by identifying and flagging dogmatic or polarizing users in online discussions. Additionally, it can be helpful in identifying personality traits of users who influence or change others' opinions, particularly regarding products or new platforms related to a specific target. This capability can aid in understanding how certain users shape discussions and change public opinion.
> >
> > *Analyzing Public Opinion Dynamics:*
> > * Researchers can use the USDC dataset to study how opinions evolve in multi-user conversations, especially in contexts like political discussions or social movements. Understanding these dynamics can provide insights into how public opinion is shaped and spread.
> >
> > *Enhancing Dialogue Systems:*
> > * The USDC dataset can be used to train dialogue systems to better understand and respond to nuanced user opinions, improving the quality of interactions in customer service, virtual assistants, and social media bots.
> > Behavioral and Social Science Research: The dataset provides a rich resource for studying behavioral patterns in online conversations, including how group dynamics influence individual opinions and the role of dogmatism in social interactions.
> >
> > *Dynamic contextual user representations:*
> > * Using USDC and user-level characteristics, we can pretrain LLMs to generate contextual embeddings specific to individual users. This approach enables the creation of dynamic user representations, allowing User-LLMs to interact more effectively with users by understanding and adapting to their unique behaviors, preferences, and communication styles.
> >
> >
> > **Q8. Rigorous Evaluation: Comprehensive evaluation of USDC dataset**
> >
> > Thank you for this interesting question.
> > * We would like to clarify that we have validated the LLM-generated annotations through several methods. We have addressed how USDC differs from prior studies and discussed its transfer learning performance on other existing datasets in our responses to CQ1 and CQ2. Since we have now completed annotations for 200 test conversations, we have reported the IAA for both humans and LLMs for the stance and dogmatism tasks in CQ3 under the common responses. Kindly also check common responses for detailed information.
> >
> > **Q9. Minor issues and typos:**
> >
> > We will correct minor issues and typos in the final version.

---

> > > ### Author Rebuttal · Authors · 2024-08-27
> > >
> > > **Q2 extension.  Individual user responses within their specific context vs. entire conversation at once for stance and dogmatism.**
> > >
> > > * Based on the reviewer's suggestion, and extending our response to Reviewer 7SSx, we conducted an additional experiment to examine the impact of recency on LLM performance for user-stance annotations.
> > >
> > > * This experiment involved verifying model annotations by focusing on the prior context for a given user, rather than considering the entire conversation.
> > > * The goal was to determine whether assessing each response within its immediate context, followed by aggregation, would yield different results compared to analyzing the full conversation context.
> > > * We now report IAA scores in **Figure 1 of the third rebuttal PDF**, which contains a matrix of Cohen's Kappa scores across different models and settings, including GPT-4 Few-Shot (FS), Mistral Large FS, Majority Voting, as well as GPT-4 FS PC and Mistral Large FS PC (which are based on prior context).
> > >
> > > **Recency vs. Full Context:**
> > >
> > > * GPT-4 FS PC and Mistral Large FS PC: These models, which are based on prior context, show different agreement levels compared to their counterparts using the full conversation context. For example:
> > > * The agreement between GPT-4 FS and Majority Voting is higher when the full conversation is considered (0.75) compared to when only prior context is used.
> > > * The agreement between GPT-4 FS PC and Mistral Large FS PC (both based on prior context) is lower than when using the full context, indicating that prior context alone may not capture all the necessary nuances for consistent annotation.
> > >
> > > **Human Agreement:**
> > >
> > > * The comparison of human annotations with models like GPT-4 FS and Mistral Large FS shows that human annotators also rely heavily on the full conversation context to maintain agreement.
> > >
> > > * The results from this additional experiment, supported by the data in **Figure 1 of the third rebuttal PDF**, suggest that while prior context can provide some useful insights, it is not as effective as considering the entire conversation context for maintaining high inter-annotator agreement.
> > >
> > > * In summary, the experiment highlights the importance of full context in LLM-based annotations and suggests that while recency can influence model performance, it should be supplemented with the entire conversation context to ensure higher accuracy and agreement.

---

> > > > ### Author Response · Authors · 2024-08-29
> > > > **Keenly awaiting your feedback**
> > > >
> > > > Dear Reviewer 7SSx,
> > > >
> > > > As we approach the end of the discussion phase, we are keen to receive your feedback on the revised manuscript. We have thoroughly addressed all the concerns you raised and conducted the suggested experiments. Additionally, we've significantly expanded the discussion on the validation of LLM-generated annotations and provided inter-annotator agreement with human labels.
> > > >
> > > > Please do let us know if there are any additional clarifications or experiments that we can offer. We would love to discuss more if any concern still remains. Otherwise, we would appreciate it if you could support the paper by increasing the score.
> > > >
> > > > regards,
> > > >
> > > > Authors

---

> > > > > ### Author Response · Authors · 2024-08-31
> > > > > **Keenly awaiting feedback on the rebuttal**
> > > > >
> > > > > Dear Reviewer 7SSx,
> > > > >
> > > > > As the author-reviewer discussion phase is nearing its conclusion, we would like to inquire if there are any remaining concerns or areas that may need further clarification. Your support during this final phase, especially if you find the revisions satisfactory, would be of great significance.  We would appreciate it if you could support the paper by increasing the score.
> > > > >
> > > > > Regards,
> > > > >
> > > > > Authors

---

### Official Review · Reviewer_KbQd · 2024-07-31

**Rating:** 7
**Confidence:** 4
**Clarity:** Yes

**Review:**

The paper presents a valuable contribution to the field of opinion analysis by introducing a novel dataset and leveraging LLMs for annotation. While the research demonstrates potential, significant limitations, particularly in evaluating annotation quality and addressing ethical concerns, hinder its overall impact.

**Strengths:**

- **Contribution**: The creation of the USDC dataset is a valuable resource for the field.
- **Novelty**: The use of LLMs for annotating such a nuanced task is innovative.
- **Potential Impact**: The automated identification of user opinions and stances has broad applications.

**Additional Feedback:**

None

**Correctness:**

Yes. But I would have like to see an evaluation of the performance of LLMs as annotators against human benchmarks on this dataset.

**Documentation:**

Yes

**Ethics:**

The paper does not raise ethical red flags but few things give me pause:

-  The paper uses Reddit conversations but it is not clear that **explicit consent** was granted to the authors to use the data in this way.
- The ability to classify user stances and dogmatism **could be misused** for manipulation purposes, such as targeted advertising or political campaigns.
- The paper **appears to implicitly trust the quality of LLM annotators** without directly comparing their performance against that of human annotators _for this task_.
- (A less immediate concern) Models trained on this dataset could potentially be **used to discriminate against people** based on their opinions or stances.

The paper does not sufficiently address any of these concerns.

**Limitations:**

While the authors have addressed several limitations, a critical gap remains in assessing annotation quality. The reliance on LLMs for data labelling without a direct comparison to human annotations precludes a definitive evaluation of the accuracy and reliability of the generated labels **for these tasks**. Inter-annotator agreement (IAA) among LLMs does not sufficiently establish the equivalence of LLM and human annotations.

**Opportunities For Improvement:**

**Rigorous Evaluation:**
Conduct a comprehensive evaluation of the dataset quality, potentially including human-in-the-loop evaluation.

**Ethical Considerations:***
Explicitly address ethical concerns related to data privacy, consent, and potential misuse of the technology.

**Paper Organization:**
The section on 'Verification using Human Interaction' should precede the 'Error Analysis' as it clarifies what standard the models are being evaluated against

**Relation To Prior Work:**

Yes

**Summary And Contributions:**

The paper takes an important step toward automating the identification of user opinions and stances in long conversation threads. The main contributions of the paper are:
- **Creating the USDC dataset**, a large-scale dataset containing multi-thread multi-user conversations on highly debated topics, likely to include opinion shifts. The dataset was annotated for user stance and user dogmatism detection.
- **Leveraging LLMs for annotation**, especially on such a nuanced task
- **Benchmarking model performance** by evaluating the performance of finetuned and instruction-tuned small language models (SLMs) on the tasks of stance and dogmatism classification, providing insights into the capabilities and limitations of these models.
- **Making the code and dataset publicly available** to foster further research in the field.

---

> ### Author Rebuttal · Authors · 2024-08-16
>
> *We thank the reviewer for their strong positive, insightful and valuable comments and suggestions which are crucial for further strengthening our manuscript.*
>
> **Q1. Rigorous Evaluation: Conduct a comprehensive evaluation of the dataset quality, potentially including human-in-the-loop evaluation.**
>
> Thank you for this important question.
> * We have addressed how USDC differs from prior studies and discussed its transfer learning performance on other existing datasets in our responses to CQ1 and CQ2. Kindly refer to those common questions for detailed information.
>
> **Q2. Ethical Considerations: Explicitly address ethical concerns related to data privacy, consent, and potential misuse of the technology.**
>
> Thank you for raising this concern.
> * While Reddit posts and comments are publicly accessible, and Reddit usernames are not real names, we want to clarify that we are not handling any personal demographic details of the users. We only consider post IDs for mapping with users, ensuring that no user identity information is revealed in our research. This approach helps to maintain user privacy while still allowing for meaningful analysis of the data.
>
> **Q3. Paper Organization: The section on 'Verification using Human Interaction' should precede the 'Error Analysis' as it clarifies what standard the models are being evaluated against**
>
> Thank you for this valuable suggestion. We will reorganize the sections as you recommended in the final version.
>
> **Q4.Limitations: Inter-annotator agreement (IAA) among LLMs does not sufficiently establish the equivalence of LLM and human annotations.**
>
> Thank you for your observation.
> * We acknowledge that inter-annotator agreement (IAA) among LLMs alone does not fully establish the equivalence of LLM and human annotations. Since we have now completed annotations for 200 test conversations, we have reported the IAA for both humans and LLMs for the stance and dogmatism tasks in CQ3 under the common responses.
> * Kindly refer to common responses questions for detailed information.

---

> > ### Author Response · Authors · 2024-08-30
> > **Keenly awaiting your feedback**
> >
> > Dear Reviewer KbQd,
> >
> > Thank you for your positive feedback and support toward the acceptance of our paper.
> >
> > As we approach the end of the discussion phase, we are eager to receive your feedback on the revised manuscript. We have thoroughly addressed all the concerns you raised and conducted the suggested experiments. Additionally, we've significantly expanded the discussion on the validation of LLM-generated annotations and provided inter-annotator agreement with human labels.
> >
> > If there are any additional clarifications or experiments that we can provide, please let us know. We would be happy to discuss further if any concerns remain.
> >
> > Best regards,
> >
> > The Authors

---

> ### Comment · Reviewer_KbQd · 2024-08-30
> **Further on IAA**
>
> Thank you for considering the comments.
>
> As I understand it, you report IAA between human annotators and LLM annotators. Can you do the same for human annotators themselves and compare that with IAA between LLM annotators? If humans agree or disagree in the same way that the LLMs do, the comparison would be more meaningful.
>
> As it stands, I'll stick with my original score.

---

> > ### Author Response · Authors · 2024-08-31
> >
> > Dear Reviewer KbQd,
> >
> > Thank you for your question. We computed the Inter-Annotator Agreement (IAA) between human annotators as well. The tables below report the IAA scores for both the stance detection and dogmatism detection tasks among the human annotators.
> >
> > **Stance Detection:**
> >
> > |                   | **Human1** | **Human2** | **Human3** |
> > |-------------------|------------------|------------------|------------------|
> > | **Human1** | 1.00             | 0.62             | 0.55             |
> > | **Human2** | 0.62             | 1.00             | 0.57             |
> > | **Human3** | 0.55             | 0.57             | 1.00             |
> >
> > **Dogmatism Identification:**
> >
> > |                   | **Human1** | **Human2** | **Human3** |
> > |-------------------|------------------|------------------|------------------|
> > | **Human1** | 1.00             | 0.57             | 0.51             |
> > | **Human2** | 0.57             | 1.00             | 0.52             |
> > | **Human3** | 0.51             | 0.52             | 1.00             |
> >
> > Summary: Based on the above tables and common response CQ3, we found that the IAA scores for stance and dogmatism are closer to the human-majority voting agreement.

---

### Author Rebuttal · Authors · 2024-08-16

*We thank the reviewers for their strong positive, insightful and valuable comments and suggestions which are crucial for further strengthening our manuscript.*

**CQ1. Rigorous Evaluation: Comprehensive evaluation of USDC dataset, including human annotations on 200 test conversations and the application of transfer learning on previously established stance datasets.**

Thank you all the reviewers for raising this concern.  We performed our evaluation on USDC dataset in two phases:

**USDC test dataset evaluation with humans labels:**
* We now completed the human annotations on 200 test samples and assessed the agreement between human labels and LLM-generated labels. We achieved an inter-annotator agreement score of 0.49 for the stance detection and 0.50 for dogmatism tasks, indicating a reasonable level of consistency between human and LLM annotations.

**Transfer Learning Performance of USDC on existing stance datasets:**

* To evaluate the quality of LLM-generated annotations, we performed transfer learning by fine-tuning the SLMs on the USDC dataset. We then tested the model’s performance on the several existing stance datasets: SPINOS, MT-CDS (suggested by reviewer w9xr) and Twitter-stance dataset.

  - **SPINOS:** Here SPINOS dataset consists of a 5-class stance detection task, as described by [Sakketou et al. 2022]. We used the same training dataset mentioned in Section 4.2. For testing, we utilized the SPINOS dataset, which consists of 3,238 post-level examples across five stance labels.
  - In comparison to the results reported by [Sakketou et al. 2022], where the best model (a traditional machine learning classifier) achieved an F1-score of 0.341, a random baseline achieved 0.230, and a majority baseline achieved 0.124, our approach using LLaMa-3-8B fine-tuning on the USDC dataset achieved a weighted F1-score of 0.320 on SPINOS. This score is close to the best model performance on the SPINOS dataset, indicating that our LLM-generated annotations on the USDC dataset are comparable in quality to human annotations.
  - **MT-CDS:** We explored the related works suggested by reviewer w9xr, highlighted the differences between our work and the suggested studies, and reported the transfer learning accuracies using the USDC dataset on these works. The MT-CSD dataset [Fuqiang Niu et al. 2024] is tailored for stance detection in multi-turn conversations with multiple targets, addressing different aspects of stance detection.
  - This dataset  consists of human annotated labels across 5 stance datasets (Biden, Bitcoin, SpaceX, Tesla, and Trump) in testing. This MT-CDS stance dataset contains 3-class labels such as favor, against and neutral. Therefore, we combined our Strongly Against and Somewhat Against as one class, Strongly In Favor and Somewhat In Favor as one class and Stance Not Inferrable as one class.  Below are the accuracies we obtained on 5 datasets (check rebuttal pdf Table 1 & 2).

|Dataset|Best Accuracy|USDC accuracy|
|----------|--------------------|---------------------|
|Biden|45.09|46.60|
|Bitcon|56.95|51.40|
|SpaceX|55.94|54.80|
|Tesla|52.38|58.3|
|Trump|48.31|60.5|
|Avg|51.73|54.32|

  - * From the above Table, we observe that our transfer learning results are closer or performing better than results reported in Table 6 of [Fuqiang Niu et al. 2024]. This implies that our LLM generated annotations are closer to human-level performance on MT-CDS stance detection dataset

  - **Twitter-stance:** This dataset focuses on extracting stance (denying vs. supporting opinions) from Twitter posts, specifically targeting replies and quotes on controversial issues. It is tailored to the specific challenges of stance detection on Twitter, particularly in controversial and rumor-related contexts. This dataset consists of 5 classes such as Implicit denial, Explicit denial, Implicit support, Explicit support, and Quotes. These classes are similar to our USDC 5-class stance labels. Below are the accuracies we obtained on twitter-stance dataset.

|Dataset|Best Micro F1-score|USDC Mirco F1-score|
|------|-------|----------|
|Twitter-stance|0.45|0.43|

   - * We also report individual class labels F1-score as follows: Denial (0.53), Support ( 0.32), Stance Not Inferrable (0.184).

   - * From Table 5 in [Villa-Cox et al. 2020], we observe that the combined quotes and replies achieve a micro F1-score of 0.45, while our approach obtained a score of 0.43, which is close to the performance of human-annotated labels. Additionally, similar to [Villa-Cox et al. 2020], our results show that the denial class performs better than the support class.

---

> ### Author Rebuttal · Authors · 2024-08-16
>
> **CQ2. How this dataset differs or is improved from previous stance classification datasets**
>
> Thank you for this important question.
> * As we discussed in the Related work section, previous studies *[Fast & Horvitz, 2016]; [Sakketou et al., 2022]*, which explore Stance and Dogmatism at the post level, where posts are randomly sampled from conversation threads. *[Fast & Horvitz, 2016]* predicted user dogmatism on randomly sampled Reddit posts from conversations, with each post limited to 200-300 characters. *[Sakketou et al. 2022]* created the post-level Stance dataset, SPINOS, where each post is considered independent, and submission posts are missing while annotators label the data.
> * Our work overcomes the limitations of previous studies and presents Stance detection for posts and Dogmatism labels of users in conversations, considering the entire context,  while preserving submission IDs. Hence, our dataset provides clear user-level posts and dogmatism data, which are useful for modeling dynamic user representations.
>
> * As per reviewer w9xr's suggestion, we will now discuss how our current USDC dataset differs from prior studies *[Fuqiang Niu et al. 2024][Villa-Cox et al. 2020][Li et al. 2023]*.
> * The USDC dataset tasks are designed to capture the nuanced shifts in user opinions over the course of a conversation.
> * In contrast, the MT-CSD dataset *[Fuqiang Niu et al. 2024]* is tailored for stance detection in multi-turn conversations with multiple targets, addressing different aspects of stance detection. Therefore, our study emphasizes tracking user opinion fluctuations, while the other study primarily focuses on multi-party discussions.
> * *[Villa-Cox et al. 2020]* work focuses on extracting stances (denying vs. supporting opinions) from Twitter posts, specifically targeting replies and quotes on controversial issues. It is tailored to the specific challenges of stance detection on Twitter, particularly in controversal and rumor-related contexts. Broadly, this work emphasizes post-level annotations in Twitter replies and quotes.
> * *[Li et al. 2023]* work focuses on target-specific stance detection, where the goal is to classify individual posts or comments into a stance class related to a specific issue, such as COVID-19 vaccination.
>
> * Overall, all prior studies contrast with the USDC dataset, which focuses on tracking user-level opinions across long, multi-user conversations, capturing the evolution of stance and dogmatism over extended discussions rather than just on a specific target issue. From the above studies, we clearly observe that these works focus more on stance detection at the post-level, while our work emphasizes user-level opinion fluctuations. Additionally, the prior studies are limited in scope, targeting specific issues (5 topics in *[Villa-Cox et al. 2020]*, 1 topic in *[Li et al. 2023]*), whereas USDC covers a broader range of general subreddits across 22 different topics.
>
> * The primary goal of USDC is to enhance personalization, market research, political campaigns, and other applications where understanding user opinions over extended discussions is crucial.
>
> *[Fuqiang Niu et al. 2024] A Challenge Dataset and Effective Models for Conversational Stance Detection*
>
> *[Villa-Cox et al. 2020] Stance in replies and quotes (srq): A new dataset for learning stance in Twitter conversations*
>
> *[Li et al. 2023] Improved target-specific stance detection on social media platforms by delving into conversation threads*
>
>
> **CQ3.  Inter-annotator agreement (IAA) among LLMs does not sufficiently establish the equivalence of LLM and human annotations.**
>
> Thank you for raising this concern.
> * We would like to clarify that we have now completed human annotation for 200 test conversations (previously, we had completed 10 sample conversations as mentioned in Section 5, Lines 320-330). We employed three human annotators to label these 200 test conversations using Google Forms, with 10 samples of conversations per form. As shown in the Figure 1 (please check rebuttal pdf), the agreement between human annotations and the few-shot versions of the models (GPT-4: few-shot, Mistral Large: few-shot) is closer to the scores of the zero-shot and one-shot configurations. Specifically, human-majority voting achieved a Cohen's Kappa score of 0.49 for stance and 0.50 for dogmatism, which aligns more closely with human annotations.
> * Following reviewers suggestion, we also calculated the inter-annotator agreement among the three human annotators themselves. The results showed an agreement of 0.57 for the stance detection task and 0.52 for the dogmatism task. These findings demonstrate the level of consistency among human annotators, providing a more comprehensive understanding of the alignment between LLM-generated labels and human judgments.

---

### Author Response · Authors · 2024-09-01
**Keenly awaiting feedback on the rebuttal**

Dear Reviewers,

As we near the conclusion of the author-discussion phase, we would like to express our sincere gratitude for the valuable comments and insights you have provided. We have carefully addressed all the concerns raised by each reviewer and conducted the suggested experiments. In addition, we have significantly expanded the discussion on the validation of LLM-generated annotations and provided detailed inter-annotator agreement with human labels.

Your feedback has been instrumental in shaping our work, and we are pleased to inform you that two reviewers now lean towards acceptance, including one rating of "7: accept, good paper." At this critical juncture, your insights and evaluation are pivotal in determining the final outcome of our work.

We would greatly appreciate your support in the form of an increased score if you find our revisions satisfactory.

Thank you once again for your time and consideration.

Best regards,
Authors

---

### Decision · Program_Chairs · 2024-09-26

**Decision:**

Reject

**Comment:**

The authors have put significant effort into building the dataset and addressing various concerns raised by reviewers. However, the dataset has several major weaknesses, including the limited additional benefits compared to existing datasets, the reliability of labels generated by LLMs, and the overall utility of the dataset. Although the authors provided human evaluation results during the rebuttal, which partially addressed some concerns, the primary issues regarding the dataset's construction and usefulness remain questionable. If we heavily rely on LLMs for label generation, the same process could be applied to a business's own dataset, which could then be used to train smaller models at a reduced cost. The authors may need to enhance this work further before it can be considered for publication.